# A neural m⁶A/Ythdf pathway is required for learning and memory in *Drosophila*

Lijuan Kan[1,7], Stanislav Ott [2,7], Brian Joseph[1,3,7], Eun Sil Park[1], Wei Dai[4], Ralph E. Kleiner[4], Adam Claridge-Chang [2,5,6] & Eric C. Lai [1✉]

Epitranscriptomic modifications can impact behavior. Here, we used *Drosophila melanogaster* to study N⁶-methyladenosine (m⁶A), the most abundant modification of mRNA. Proteomic and functional analyses confirm its nuclear (Ythdc1) and cytoplasmic (Ythdf) YTH domain proteins as major m⁶A binders. Assays of short term memory in m⁶A mutants reveal neural-autonomous requirements of m⁶A writers working via Ythdf, but not Ythdc1. Furthermore, m⁶A/Ythdf operate specifically via the mushroom body, the center for associative learning. We map m⁶A from wild-type and *Mettl3* mutant heads, allowing robust discrimination of Mettl3-dependent m⁶A sites that are highly enriched in 5′ UTRs. Genomic analyses indicate that *Drosophila* m⁶A is preferentially deposited on genes with low translational efficiency and that m⁶A does not affect RNA stability. Nevertheless, functional tests indicate a role for m⁶A/Ythdf in translational activation. Altogether, our molecular genetic analyses and tissue-specific m⁶A maps reveal selective behavioral and regulatory defects for the *Drosophila* Mettl3/Ythdf pathway.

¹Department of Developmental Biology, Sloan-Kettering Institute, New York, NY, USA. ²Program in Neuroscience and Behavioral Disorders, Duke-NUS Medical School, Singapore, Singapore. ³Louis V. Gerstner, Jr. Graduate School of Biomedical Sciences, Memorial Sloan Kettering Cancer Center, New York, NY, USA. ⁴Department of Chemistry, Princeton University, Princeton, NJ, USA. ⁵Institute for Molecular and Cell Biology, A*STAR, Singapore, Singapore. ⁶Department of Physiology, National University of Singapore, Singapore, Singapore. ⁷These authors contributed equally: Lijuan Kan, Stanislav Ott, Brian Joseph. ✉email: laie@mskcc.org

About 45 years ago, pioneering studies led by Seymour Benzer identified *dunce*, the first learning mutant in any animal[1], and established *Drosophila* as an important model to elucidate mechanisms of learning and memory[2]. Although flies execute a broad repertoire of learned behaviors[3,4], associative odor learning remains the most widely studied type of learning in this organism[5]. In the aversive olfactory-conditioning paradigm, flies are presented with a pair of neutral odors in succession, one in the presence of electric shock and the other without. Subsequently, when flies encounter this odor again later in the absence of shock, the shock-paired odor elicits an avoidance response. A single training trial is sufficient to induce short-term odor-avoidance memory, which can last several hours[6]. Studies during the past decades identified dozens of protein-coding genes[5] and a half-dozen microRNAs[7] that are required for normal short-term memory (STM) formation.

Along with advances in *Drosophila* memory genetics, substantial progress has been made in deciphering the neuronal anatomy and circuits that underlie memory[4,8]. The mushroom bodies (MB) have been revealed as the higher-order brain center for associative learning[9]; intrinsic MB neurons called Kenyon cells (KC)[10] receive olfactory signals from primary olfactory sensory neurons[11,12] via the antennal lobe[13,14]. As in some vertebrate systems[15], neuronal ensembles in the *Drosophila* MB are thought to represent odor-memory engrams that are continuously modified by the animal's experience. More recent studies provided a higher functional resolution of MB compartments (lobes) with regards to various types of olfactory memory[8]. Moreover, clusters of dopaminergic neurons, such as PPL1[16] and PAM[17,18] were found to innervate distinct MB lobes and provide instructive value to the perceived olfactory stimulus.

In our effort to identify additional factors that regulate memory, we were enticed by the "epitranscriptome", the multitude of modified bases that exist beyond the standard RNA nucleotides. The most abundant and most well-studied internal modification of mRNA is $N^6$-methyladenosine (m6A)[19]. While m6A has been recognized to exist in mRNA since the 1970s[20,21], its functional significance has been elusive until recently. Key advances included (1) techniques to determine individual methylated transcripts, and in particular specific methylated sites, and (2) mechanistic knowledge of factors that install m6A ("writers") and mediate their regulatory consequences ("readers"). The core m6A methyltransferase complex acting on mRNA consists of the Mettl3 catalytic subunit and its heterodimeric partner Mettl14. These associate with other proteins that play broader roles in splicing, mRNA processing and gene regulation, but that are collectively required for normal accumulation of m6A[19].

Downstream of the writers, various readers are sensitive to the presence or absence of m6A, and thereby mediate differential regulation by this mRNA modification[22]. The most well-characterized readers contain YTH domains, for which atomic insights reveal how a tryptophan-lined pocket selectively binds methylated adenosine and discriminates against unmodified adenosine[23–26]. In addition, some other proteins were proposed as m6A readers, based primarily on preferential in vitro binding to methylated vs. unmethylated RNA probes. In mammals, m6A readers confer diverse regulatory fates onto modified transcripts, including splicing[27] and nuclear export[28] via the nuclear reader YTHDC1, and RNA decay via cytoplasmic readers Ythdf1-3[29–32]. Certain YTHDF[33–37] and YTHDC2[38] were also reported to regulate translation via m6A under specific contexts.

Despite intense efforts into m6A mechanisms and genomics using cell systems, genetic analyses of the m6A pathway have only begun in earnest in the past few years, mostly in vertebrates. Notably, many studies have revealed sensitivity of the mammalian nervous system to manipulation of m6A factors[39]. Mutants in writer (*Mettl3* and *Mettl14*), reader (primarily *ythdf1*), and eraser (*FTO*) factors have collectively been shown to exhibit aberrant neurogenesis and/or differentiation[40–45]. Moreover, these mutants impact neural function and behavior, including during learning and memory paradigms[34,46–50]. Overall, these observations may reflect some heightened requirements for m6A in neurons, perhaps owing to their unique architectures and/or regulatory needs.

Amongst invertebrates, *Caenorhabditis elegans* lacks the core m6A machinery[51], but the presence of a *Drosophila* ortholog of Mettl3 (originally referred to as IME4) opened this model system[52]. While mammals contain multiple members of both nuclear and cytoplasmic YTH domain families, the fly system is simplified in containing only one of each, referred to as Ythdc1 (YT-521B or CG12076) and Ythdf (CG6422), respectively. Recently, the Soller, Roignant and Lai labs established biochemical, genetic, and genomic foundations for studying the m6A pathway in *Drosophila*[53–55]. Surprisingly, these studies jointly reported that knockout of all core m6A writer factors in *Drosophila* is compatible with viability and largely normal exterior patterning. Nevertheless, mutants of *Mettl3*, *Mettl14*, and *Ythdc1* exhibit a common suite of molecular and phenotypic defects. These include several behavioral abnormalities as well as aberrant splicing of the master female sex determination factor *Sex lethal* (*Sxl*). The suite of locomotor and postural defects in *Drosophila* m6A mutants was again consistent with the notion that the nervous system might be especially sensitive.

However, a major open question from these studies concerns the regulatory and biological roles of the sole *Drosophila* cytoplasmic YTH factor, Ythdf. In contrast to other core m6A factors, we did not previously observe overt defects in our *Ythdf* mutants, nor did it seem to exhibit robust m6A-specific binding activity[55]. Here, we use proteomic analyses to reveal Ythdc1 and Ythdf as the major m6A-specific binders in *Drosophila*, and focused biochemical tests show that Ythdf prefers a distinct sequence context than tested previously. Hypothesizing that the nervous system might exhibit particular needs for the m6A pathway, we utilized a paradigm of aversive olfactory conditioning to reveal an m6A/ Ythdf pathway that is important for STM in older animals. We complement these phenotypic data with high-stringency maps of methylated transcript sites from fly heads, and show that m6A does not impact transcript levels but is preferentially deposited on genes with lower translational efficiency. Nevertheless, functional tests reveal that Mettl3/Ythdf can enhance protein output. Finally, we show that physiological Mettl3/Ythdf function is explicitly required within mushroom body neurons to mediate normal conditioned odor memory during aging. Overall, our study provides insights into the in vivo function of this mRNA modification pathway for normal behavior.

## Results

**Drosophila Ythdc1 and Ythdf bind m6A in A-rich contexts.** In mammals, two general classes of m6A-binding proteins ("readers") are recognized, based on whether they contain or lack a YTH domain[22]. Although evidence has been shown for preferential association to m6A vs. A for non-YTH proteins, the YTH domain is the only module for which the structural basis of selective m6A binding is known.

The *Drosophila* genome encodes single orthologs of nuclear (Ythdc1) and cytoplasmic (Ythdf) YTH factors. We previously tested capacities of their isolated YTH domains to associate preferentially with m6A, using RNA probes bearing GGm6ACU vs. GGACU contexts[55]. This motif represents the favored binding site for mammalian YTHDC1, which has explicitly been shown to prefer G and disfavor A at the −1 position[26]. Of note, however,

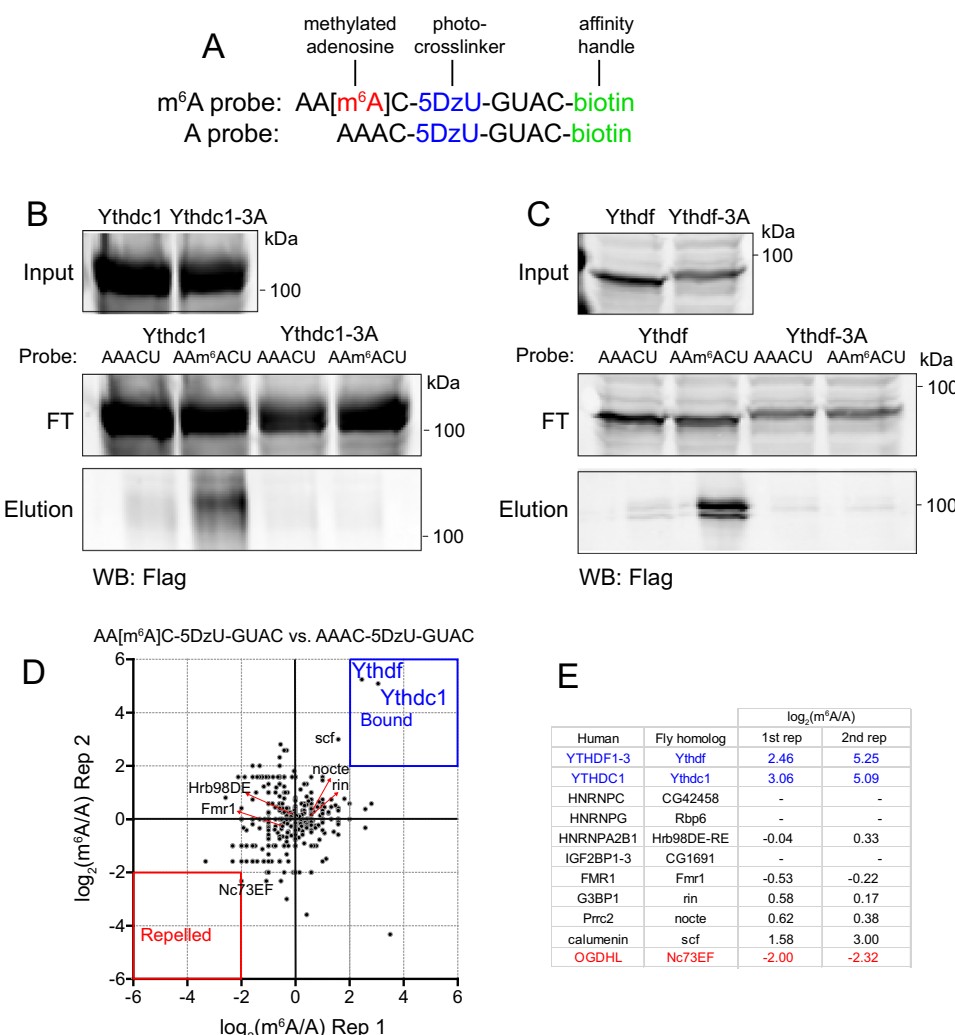

**Fig. 1 Ythdc1 and Ythdf are the major m⁶A-binding proteins in *Drosophila*. A** RNA photoaffinity probes used in crosslinking assays. **B** and **C** Both nuclear (Ythdc1) and cytoplasmic (Ythdf) *Drosophila* YTH factors specifically recognize m⁶A within the AAm⁶ACU context. Point mutations of aromatic residues that line the m⁶A cage (3A variants) abolish selective binding to m⁶A probes. FT Flow through. Shown is one representative result from two repeats. **D** Proteomic profiling of S2 cells using m⁶A and A RNA photoaffinity probes reveals Ythdc1 and Ythdf as the only preferentially bound (reader) proteins, and no strongly repelled proteins were found in these conditions. Background proteins are clustered together around the plot origin; threshold = 2X the interquartile range. Two biological replicates were plotted. **E** Selected values of fly homologs of mammalian m⁶A readers and repelled proteins expressed in S2 cells, as well as candidates of novel bound/repelled factors. Source data are provided as a Source Data file and Supplementary Dataset 1.

mammalian YTHDF1 does not share this discriminatory feature[25,56]. We previously observed the YTH domain of *Drosophila* Ythdc1 exhibits robust and selective binding to this methylated probe, but the corresponding domain of Ythdf had only modest activity. From these tests, it was not clear whether the isolated YTH domain might not be fully functional, or perhaps prefers a distinct target site. We tested both of these notions.

We compared the binding of full-length Ythdc1 and Ythdf proteins to m⁶A vs. A using biotinylated RNA photoaffinity probes[57]. These probes contain diazirine-modified uridine (5-DzU) that can be cross-linked to protein upon UV irradiation (Fig. 1A). We have shown that 5-DzU does not interfere with protein binding at the modified nucleotide, and therefore enables high-efficiency detection of associated proteins[57]. We incubated cell lysates expressing tagged YTH proteins with beads conjugated to GGm⁶ACU/GGACU RNA probes, immunoprecipitated complexes with streptavidin, and performed Western blotting for YTH factors. We observed modestly enhanced association of Ythdc1 to GGm⁶ACU vs. GGACU, while Ythdf did not crosslink preferentially to this methylated probe (Supplementary Fig. 1).

As our previous mapping suggested that *Drosophila* m⁶A modifications are biased to have upstream adenosines[55], we next compared AAm⁶ACU/AAACU probes. Interestingly, both Ythdc1 and Ythdf exhibited clearly preferential binding to methylated adenosine in this context (Fig. 1B, C). Next, we tested variants in which three critical tryptophan/leucine residues in the m⁶A-binding pocket were mutated to alanine (Supplementary Fig. 1). Although "3A" mutant proteins accumulated to similar levels as their wild type counterparts, both Ythdc1-3A and Ythdf-3A failed to bind m⁶A (Fig. 1B, C), indicating that their specificity for methylated RNA requires intact YTH domains. Thus, *Drosophila* YTH proteins, in particular Ythdf, may prefer an A-rich context.

**Ythdc1 and Ythdf are the dominant *Drosophila* proteins specifically bound to AAm⁶ACU probes.** Having clarified that both fly YTH factors specifically discriminate between m⁶A and A, we sought to identify differential binders using an unbiased approach. Proteomic studies in mammalian cells reveal YTH factors as dominant proteins that preferentially associate with

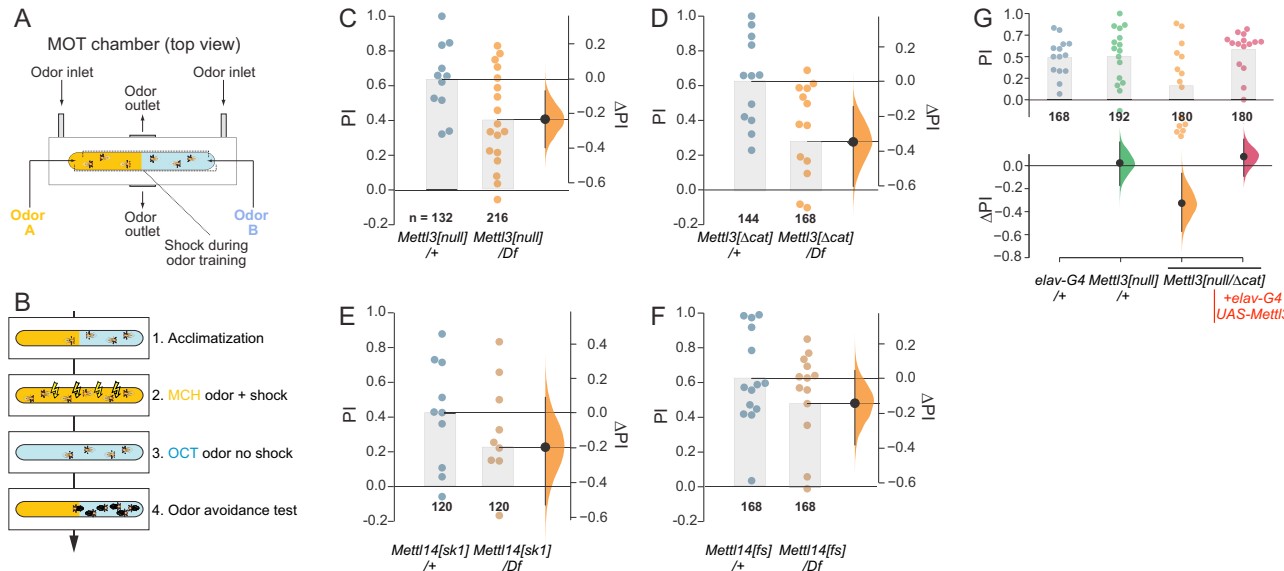

**Fig. 2 m⁶A pathway is required for short-term learning and memory (STM) in *Drosophila*. A** Schematic of the multifly olfactory training (MOT) chamber apparatus used for behavioral measurements. **B** Paradigm for shock-associated odor avoidance assay for STM acquisition. **C** and **D** Hemizygote null conditions for *Mettl3[null]* and *Mettl3[Δcat]* both led to STM impairment. *Mettl3[null]/Df* (n = 216) vs. *Mettl3[null]/+* (n = 132) = −0.23[95CI −0.4, −0.06] p = 0.033. *Mettl3[Δcat]/Df* (n = 168) vs. *Mettl3[Δcat]/+* (n = 144) = −0.35[95CI −0.6, −0.14] p = 0.017. **E** and **F** Hemizygote null conditions for *Mettl14[fs]* and *Mettl14[sk1]* led to a mild STM impairment. *Mettl14[sk1]/Df* (n = 120) vs. *Mettl14[sk1]/+* (n = 120) = −0.19[95CI −0.52, +0.14] p = 0.345. *Mettl14[fs]/Df* (n = 168) vs. *Mettl14[fs]/+* (n = 168) = −0.15[95CI −0.37, +0.06] p = 0.505. **G** Pan-neuronal expression of *UAS-Mettl3* using *elav-Gal4* (*elav-G4*) in *Mettl3[null]* hemizygotes rescued their STM phenotype. *Mettl3[null]/+* (n = 192) vs. *elav-G4/w^{1118}* (n = 168) = +0.02[95CI −0.17, +0.21] p = 0.647. *Mettl3[null]/Mettl3[Δcat]* (n = 180) vs. *elav-G4/w[1118]* (n = 168) = −0.33[95CI −0.57, −0.07] p = 0.085. *Mettl3[null]/Mettl3[Δcat]* + *elav-G4 UAS-Mettl3* (n = 180) vs. *elav-G4/w[1118]* (n = 168) = +0.08[95CI −0.1, +0.22] p = 0.132. All assays in **C–F** were conducted in 20-day-old flies. Bars present mean values. Each dot in the scatter plots represents a PI average of 12 flies. All control–test differences are displayed as effect sizes with error curves and 95% confidence intervals. No null-hypothesis significance testing was performed; two-tailed Mann–Whitney *P* values are shown for legacy purposes only. For multiple comparisons, several test groups were compared against a common control group. PI = performance index. Source data are provided as a Source Data file.

m⁶A compared to unmethylated probes, along with some other proteins (e.g. FMRP and LRPPRC), and reciprocally some factors that are repelled by this modification (e.g. stress granule factors such as G3BP1/2, USP10, CAPRIN1, and RBM42)[57,58]. As well, other methods were used to identify mammalian factors that appear to bind preferentially to m⁶A, such as Prrc2a[59] and IGF2BP1-3[60].

We used our AAm⁶ACU/AAACU RNA photoaffinity probes to pull down endogenous proteins from S2 cell lysates, followed by mass spectrometry. We performed replicate proteomic assays, and plotted the ratios of peptide counts recovered from m⁶A and A probes (Fig. 1D and Supplementary Dataset 1). These experiments revealed Ythdc1 and Ythdf were strongly and reproducibly enriched with the m⁶A probe compared to the A probe. By contrast, we did not observe clearly differential association of any other factors, including all fly homologs of other mammalian proteins reported to preferentially bind or be repelled by m⁶A[22] (Fig. 1D, E).

Overall, while conceivable that other target sequences or lysate sources might reveal additional differential binders, we subsequently focused on YTH domain factors as the major direct readers for m⁶A biology in *Drosophila*.

**Neural autonomous function of m⁶A supports olfactory learning**. The expression of several m⁶A factors is elevated in the *Drosophila* nervous system, and mutants of m⁶A factors are viable, but some exhibit locomotor defects[53–55]. As this suggested preferential sensitivity of the nervous system to m⁶A, we examined phenotypic requirements of neural m⁶A in greater detail.

Recent studies reported that the m⁶A pathway is required for learning and memory in mice[34,46]. To investigate whether this is also true for flies, we used a classical aversive conditioning paradigm to test *Drosophila* m⁶A mutants for deficits in STM. To obtain time-resolved performance measurements, we employed a conditioning apparatus that we named the multi-fly olfactory trainer (MOT, Fig. 2A and Supplementary Fig. 2). Briefly, during olfactory training one odor is administered in the presence of a shock stimulus while the other odor is subsequently delivered in the absence of foot shock (Fig. 2B). Because shock is innately aversive, *Drosophila* will associate the odor given in the presence of shock with harm and will tend to avoid it during later encounters. During the test phase the flies are presented with both odors and the avoidance of the conditioned odor is quantified as a measure of aversive shock-odor memory.

We used the respective heterozygotes as controls in following tests. To minimize background genetic effects, a frequent confound of behavioral assays, we compared these to trans-heterozygous or hemizygous (over deficiency) allelic combinations. In young flies, both writer mutants were essentially normal: we observed only a modest STM reduction in 10-day-old *Mettl3* hemizygous nulls, while similarly aged *Mettl14* mutants showed no impairment (Supplementary Fig. 3). However, at 20 days, both *Mettl3[null]* and *Mettl3[Δcat]* hemizygous nulls displayed a substantially stronger (ΔPI −0.2 to −0.3) STM impairment (Fig. 2C, D). Consistent with the role of Mettl14 as a cofactor for Mettl3, hemizygote *Mettl14[fs]*, and *Mettl14[SK1]* mutants also exhibited comparable STM impairments in 20-day-old flies (Fig. 2E, F).

Assays of whole animal mutants did not resolve if the nervous system per se was involved in these behavioral defects. We

addressed this using tissue-specific knockdown and rescue experiments. We first generated *Mettl3[null]* hemizygote animals bearing *elav-Gal4* and *UAS-Mettl3* transgenes, to drive their expression in all neurons. In this genetic background, all non-neuronal cells of the intact animal lack Mettl3. Strikingly, these flies exhibited normal STM (Fig. 2G), providing stringent evidence that the odor avoidance behavioral defect of m⁶A knockouts is strictly due to a cell-autonomous function of Mettl3 in neurons.

**Ythdf, but not Ythdc1, is the functional effector of m⁶A during STM.** We sought to elaborate the regulatory pathway underlying m⁶A in learning and memory. Prior genetic assays linked m⁶A writers Mettl3/Mettl14 in a pathway with nuclear reader Ythdc1 for locomotor and gravitaxis behaviors, as well as ovary development[53–55]. By contrast, our *Ythdf* mutants did not resemble other core m⁶A mutants, and overall seemed to lack substantial defects in these assays[55].

The phenotypic discrepancy of these mutants was further emphasized by quantifying their lifespans. While mutations in *Mettl3* and *Ythdc1* led to severely shortened lifespan (>40 days), loss of *Ythdf* had only minor effects on lifespan (Fig. 3A–C and Supplementary Fig. 4a–c). As some behavioral effects of the m⁶A pathway are mediated by the nervous system[54], we tested the effect of neuronal loss of *Mettl3*. After validating that a *UAS-Mettl3-RNAi* transgene was able to deplete both *Mettl3* RNA and protein (Supplementary Fig. 5a, b), we tested the consequences of pan-neuronal depletion using *elav-Gal4*. Compared to controls, loss of neural *Mettl3* caused modestly shorter (10 days) lifespan (Supplementary Fig. 4d). These data support the concept of a physiologically important role for nuclear readout of m⁶A via Ythdc1, with overt, nervous-system effects on longevity.

In light of extensive locomotor defects and short lifespan of *Ythdc1* mutants, we were surprised to find that *Ythdc1* nulls had normal STM performance at 10 and 20 days of age (Supplementary Fig. 3D and Fig. 3D). We also depleted *Ythdc1* using RNAi (Supplementary Fig. 5a), but pan-neuronal knockdown of *Ythdc1* using *elav-Gal4* in aged flies also did not affect STM (Fig. 3E). Thus, we were prompted to examine mutants of the cytoplasmic reader YTHDF more carefully. Excitingly, *Ythdf* hemizygotes exhibited age-related STM impairment (Fig. 3F), comparable to *Mettl3* mutants. Since *Ythdc1* mutants generally phenocopy other defects of m⁶A writer mutants, these data indicate a division of labor between the *Drosophila* YTH readers, downstream of m⁶A writers.

To test whether Ythdf was specifically required in the nervous system, we used a validated RNAi transgene (Supplementary Fig. 5a). 20-day-old *elav-Gal4 > UAS-Ythdf[RNAi]* flies also exhibited impaired STM (Fig. 3G). Altogether, these data indicate that cytoplasmic readout of m⁶A by Ythdf is required for normal function of memory-storing neurons in older flies.

**Normal locomotor response and olfactory acuity in *Mettl3* and *Ythdf* mutants.** Since STM performance depends on locomotion and olfaction, we asked if the defects in *Mettl3* and *Ythdf* mutants were specific to memory-related behavior. We note that a previous study reported adult walking defects in m⁶A mutants[54]; however, this was measured using a different assay in which the fly wings are removed and animals are provided visual landmarks to promote directional movements (Buridan's paradigm). We examined the locomotor response of *w[1118]* and m⁶A mutants before, during, and after a shock stimulus in the MOT. Both *Mettl3* and *Ythdf* locomotor responses to shock were very similar to controls (Supplementary Fig. 6a), indicating that m⁶A mutants

do not exhibit shock sensitivity or locomotion defects in the MOT setup that might affect the memory measurements.

Second, we tested the olfactory acuity of aged m⁶A mutants. Given a choice between the MCH odor and clean air (in the absence of conditioning), *w[1118]*, *Mettl3*, and *Ythdf* avoidance scores were all comparable (Supplementary Fig. 6b, c). This was true for both the same MCH concentration used in conditioning experiments, and a four-fold higher concentration. Thus, the olfactory acuity of m⁶A mutants appears normal.

**Neither Mettl3 nor Ythdf can cross-rescue each other's memory defects.** We next asked whether overexpression of Ythdf in *Mettl3* mutants, or the reciprocal genetic manipulation, would affect STM. Successful rescue could, for example, suggest that the reading function of Ythdf is not fully dependent on Mettl3 methylation, i.e. may somehow involve a parallel pathway. However, *Mettl3* nulls supplemented with pan-neuronal Ythdf overexpression did not show STM improvement (Supplementary Fig. 7a). Similarly, overexpression of Mettl3 did not improve the STM impairment in *Ythdf* mutants (Supplementary Fig. 7b). Beyond serving as stringent negative controls for the cognate rescue experiments (Figs. 2 and 3), these results provide further credence to the notion that a linear, directional Mettl3/14 → Ythdf pathway underlies m⁶A-mediated function for STM.

**Mapping the Mettl3-dependent m⁶A methylome in *Drosophila*.** To link these brain-function defects to the underlying molecular landscape of RNA methylation, we sequenced m⁶A sites from polyadenylated transcripts using miCLIP[61]. Although we previously reported miCLIP datasets from *Drosophila* embryos[55], we recognized that there can be background association in such data. Thus, individual sequencing "peaks" need to be interpreted cautiously. To provide a stringent basis to infer the existence of m⁶A at given sites, we analyzed companion input and miCLIP libraries from dissected heads, which are highly enriched for neurons, comparing wild-type and deletion mutants of *Mettl3*, which encodes the catalytic methyltransferase subunit essential for mRNA modification (e.g. Fig. 1 and Supplementary Dataset 2).

The miCLIP libraries from *Mettl3* mutants proved especially valuable, because they allowed us to distinguish m⁶A-IP loci that were clearly genetically dependent on endogenous Mettl3 (Fig. 4A, Supplementary Fig. 8a). Reciprocally, numerous regions of the transcriptome were significantly enriched in miCLIP libraries compared to input, but whose signals persisted in *Mettl3* mutants (Fig. 4B, Supplementary Fig. 8b). These might conceivably represent transcript regions modified by another factor[62], but cannot at this point be easily distinguished from non-specific pulldown. In general, the Mettl3-independent peaks were globally present in weaker m⁶A peaks (Fig. 4C), suggesting they are functionally less relevant. Therefore, we applied stringent filtering to focus our attention on the rich set of clearly Mettl3-dependent peaks (Fig. 4A–C). In addition, as we employed strong selection for polyadenylated transcripts for input, we prioritized studies of annotated genes. Altogether, our analyses (see the "Methods" section) yielded 3874 Mettl3-dependent peaks from 1635 genes. Since a subset of these called regions contained clear local minima, we applied PeakSplitter[63] to arrive at 4686 head m⁶A peaks (Supplementary Dataset 3).

***Drosophila* m⁶A is highly enriched in 5′ UTRs within adenosine-rich contexts.** Characterization of *Drosophila* Mettl3-dependent m⁶A peaks revealed fundamental similarities and differences with m⁶A patterns in other organisms. Mammalian (e.g. human and mouse) m⁶A is well-known to dominate at stop codons and 3′ UTRs[19,64]. In fish, m⁶A is also highly enriched at

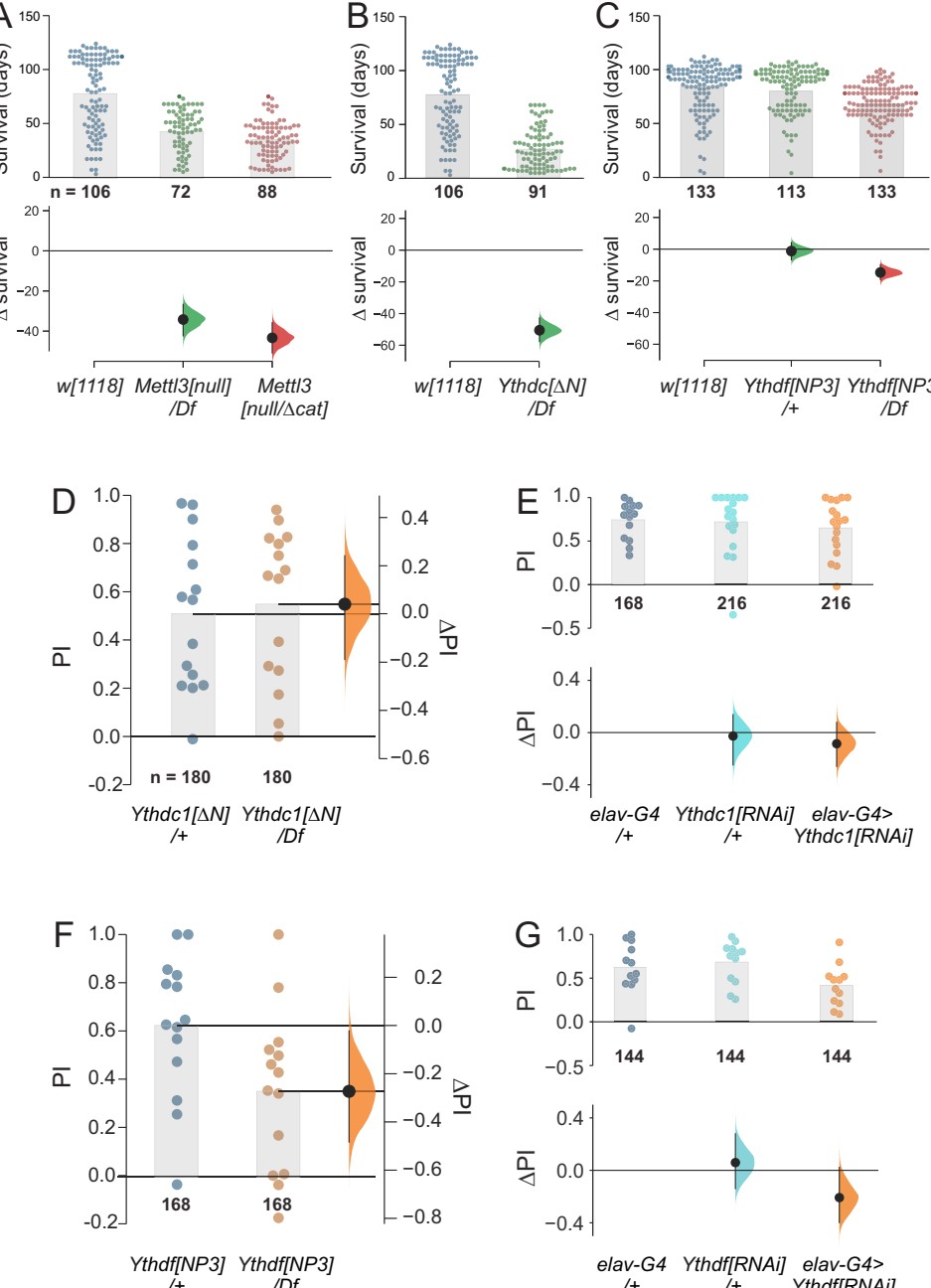

**Fig. 3 Ythdf, but not Ythdc1, mediates the role of m6A in *Drosophila* STM. A–C** Lifespan measurements of m6A writer and reader mutants. Mutants of *Mettl3* (**A**) and nuclear reader *Ythdc1* (**B**) exhibit severely shortened lifespan, but mutants of cytoplasmic reader *Ythdf* (**C**) shows only a minor lifespan reduction. *Mettl3[null]/Df* (n = 72) vs. *w[1118]* (n = 106) = −34.2[95CI −42.1, −26.31] p < 1*10^−4. *Mettl3[null]/*Mettl3[Δcat] (n = 88) vs. *w[1118]* (n = 106) = −43.3[95CI −50.66, −35.82] p < 1*10^−4. *Ythdc[ΔN]/Df* (n = 91) vs. *w[1118]* (n = 106) = −50.5[95CI −57.53, −42.81] p < 1*10^−4. *Ythdf[NP3]/+* (n = 113) vs. *w[1118]* (n = 133) = −1.2[95CI −6.57, +4.24] p = 0.431. *Ythdf[NP3]/Df* (n = 133) vs. *w[1118]* (n = 133) = −14.7[95CI −19.37, −9.96] p < 1*10^−4. **D–G** STM measurements in 20-day flies. **D** Despite gross behavioral defects and short lifespan, *Ythdc1* mutants exhibit normal STM. *Ythdc[ΔN]/Df* (n = 180) vs. *Ythdc1[ΔN]/+* (n = 180) = +0.04[95CI −0.18, +0.25] p = 0.709. **E** Pan-neuronal knockdown of *Ythdc1* using *elav-Gal4* (*elav-G4*) also yields normal STM. *Ythdc1[RNAi]/+* (n = 216) vs. *elav-G4/+* (n = 168) = −0.03[95CI −0.25, +0.14] p = 0.717. *Elav-G4/Ythdc1[RNAi]* (n = 216) vs. *elav-G4/+* (n = 168) = −0.09[95CI −0.26, +0.08] p = 0.568. **F** *Ythdf* hemizygotes recapitulate age-induced STM impairment seen in m6A writer mutants. *Ythdf [NP3]/Df* (n = 168) vs. *Ythdf[NP3]/+* (n = 168) = −0.27[95CI −0.49, −0.04] p = 0.023. **G** Pan-neuronal knockdown of *Ythdf* compromised STM similar to the impairment observed in whole animal *Ythdf* mutants. *Ythdf[RNAi]/+* (n = 144) vs. *elav-G4/+* (n = 144) = +0.06[95CI −0.14, +0.28] p = 0.707. *Elav-G4/Ythdf[RNAi]* (n = 144) vs. *elav-G4/+* (n = 144) = −0.21[95CI −0.4, +0.02] p = 0.053. STM assays in **D–G** were conducted in 20-day-old flies. Bars in D–G represent mean values. All control–test differences are displayed as effect sizes with error curves and 95% confidence intervals, two-tailed Mann–Whitney P values are shown for legacy purposes only. Source data are provided as a Source Data file.

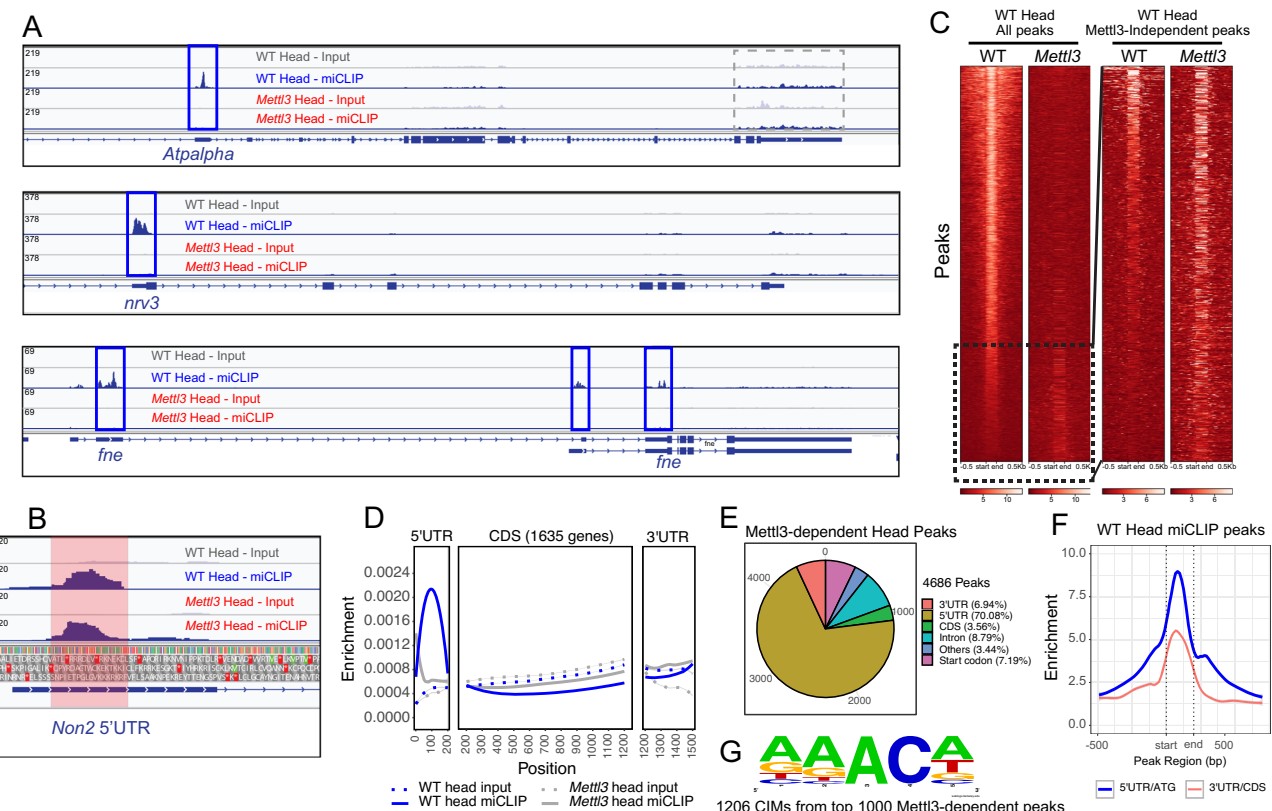

**Fig. 4 High stringency mapping of the *Drosophila* m⁶A methylome reveals new features. A** IGV screenshots of genes that exemplify archetypal 5′ UTR miCLIP enrichment and the utility of wild type vs. *Mettl3* mutant comparisons. The IGV tracks above the transcript model depict miCLIP and input libraries from wild type and *Mettl3* female heads. Note the 5′ UTRs of all three genes (*Atpalpha, nrv3*, and *fne*) contain prominent Mettl3-dependent peaks (blue box). Other exonic regions sequenced in miCLIP libraries are not enriched above input libaries (e.g. gray box). **B** IGV screenshot of *Non2* illustrates a 5′UTR Mettl3-independent peak (red box). **C** Heatmaps of Mettl3-dependent and -independent m⁶A peaks. Heatmaps of input normalized miCLIP signals at Mettl3-dependent and independent peaks. Each line represents miCLIP enrichment over input across a MACS2-called peak, as well as 500 nt flanking regions. Heatmaps on the left are all peaks including Mettl3-dependent and independent m⁶A peaks. Mettl3-independent m⁶A peaks are displayed in the right panels. Note that the scales are different for the heatmap panels. Approximate locations of Mettl3-independent m⁶A peaks within all peaks are indicated by a dashed box. **D** Metagene profiles of miCLIP and input signals along a normalized transcript using genes that contain high-confidence m⁶A peaks in head libraries. An overwhelming 5′ UTR enrichment is observed. **E** Metagenes of enrichment at high-confidence m6A peaks that have been group as 5′ UTR/start codon and 3′ UTR/CDS regions. Metagenes are produced by averaging signals from input normalized miCLIP from WT head. On average, stronger signals are observed at 5′ UTRs/start codons. Peak start and end are specified on the *x*-axis. Dashed lines include the start and end locations of peaks. **F** Pie chart depicting the fraction of m⁶A peaks in different transcript segments. **G** Nucleotide content surrounding CIMs located within the top 1000 Mettl3-dependent m⁶A peaks in head. Source data are provided as a Source Data file and Supplementary Dataset 3.

stop codons, but the predominant Mettl3-dependent signals localize to 5′ UTRs[65]. Previous work in *Drosophila* was conflicting, since low-resolution meRIP-seq suggested mostly CDS modification with a small minority in UTRs[54], while our prior miCLIP data indicate dominant UTR modifications, preferentially in 5′ UTRs[55]. However, these maps were generated with different technologies, and neither was controlled against mutants.

Our miCLIP data provide a clearer perspective. Strikingly, while found at some level throughout the transcriptome, m⁶A predominates in 5′ UTRs in *Drosophila*. This can be observed at numerous individual loci (Fig. 4A and Supplementary Fig. 8) and via miCLIP metagene profiles (Fig. 4D). Overall, while we do observe some Mettl3-dependent coding sequence (CDS) and 3′ UTR miCLIP peaks (Fig. 4E, and Supplementary Fig. 8), these were overall rare, of generally lower ranks than 5′ UTR and start codon peaks (Fig. 4F), and not appreciably enriched in metagene profiles over companion mutant datasets (Fig. 4D).

We examined C-to-T crosslinking-induced mutations following adenosine residues (CIMs), which have been taken to represent individual m⁶A site in miCLIP data[61,66]. In particular,

we focused on CIMs located within Mettl3-dependent m⁶A-IP peaks, which we took as bearing high-confidence RNA methylation sites (Supplementary Dataset 4). Within these, the sequence context of CIMs in *Drosophila* roughly resembles the DRAC context that has been observed in other species[19]. However, while a majority of sites fall into a GGACH context in vertebrates[19], m⁶A sites in *Drosophila* prefer AAACD (Fig. 4G), correlating with the preferred binding sites of Ythdc1 and Ythdf in our assays of photocrosslinking-activated m⁶A probes (Fig. 1).

We validated our map by testing m⁶A-IP to IgG-IP samples for enrichment of m⁶A target transcripts using rt-qPCR (Supplementary Fig. 9). We validated a number of top m⁶A targets (e.g. *aqz, Syx1A, fl(2)d, prosap, pum, futsch, gish*) from whole female fly RNA (Fig. 5A). Still, recognizing that m⁶A-RIP-qPCR evaluates the presence of entire transcripts in pulldowns, we performed parallel experiments from *Mettl3[null]* female flies. All of these binding events, even loci with very modest enrichment in wild type (e.g. *sky*, Fig. 5A), were found to be Mettl3-dependent. By contrast, control loci lacking m⁶A peaks (*fwe* and *CG7970*) showed very little m⁶A-dependent IP signals, and these were

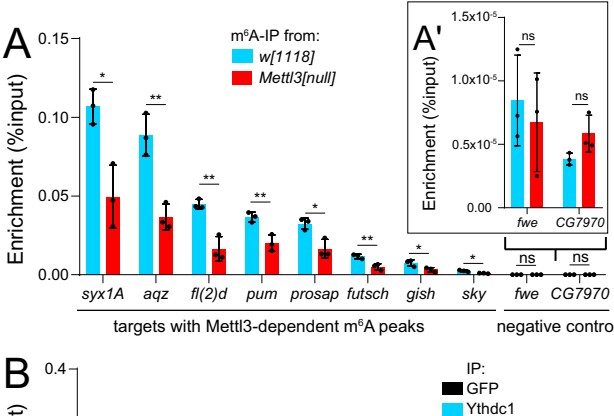

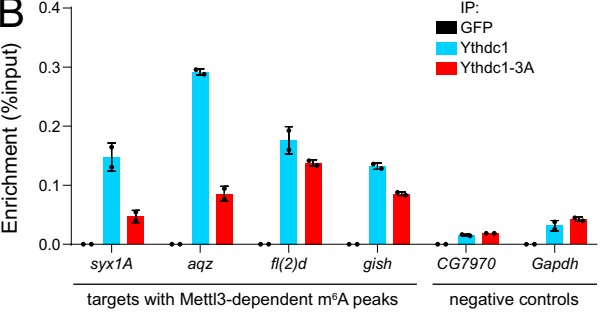

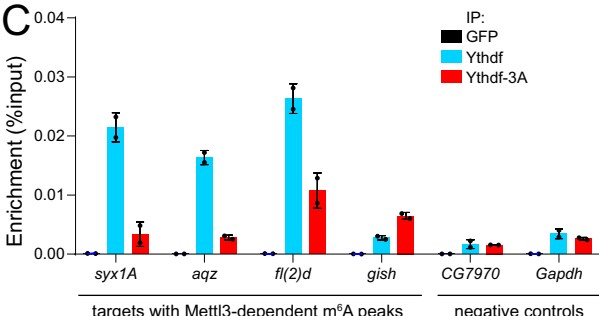

**Fig. 5 Validation of *Drosophila* m⁶A targets and their association with readers. A–A′** m⁶A-RIP-qPCR validation of m⁶A transcripts in *w[1118]* control and *Mettl3[null]* knockout whole female flies. Error bars, mean ± SD; *n* = 3 biological replicates. Two-tailed *t*-test, **p* < 0.05, ***p* < 0.01. *syx1A* *p* = 0.012, *aqz* *p* = 0.005, *fl(2)d* *p* = 0.004, *pum* *p* = 0.009, *prosap* *p* = 0.016, *futsch* *p* = 0.009, *gish* *p* = 0.044, *sky* *p* = 0.010. n.s. non-significant, *fwe* *p* = 0.604, *CG7970* *p* = 0.085. **B** and **C** RIP-qPCR of m⁶A targets in S2-S cells and transfected Ythdc1 (**B**) or Ythdf (**C**) constructs shows specific pulldown of several targets relative to GFP-RIP control, and the association of several of these is compromised by mutation of the YTH domain (3A versions). Shown is one representative result from three independent repeats. Source data are provided as a Source Data file.

the test proteins and target RNAs, which may or may not occur directly through the modified nucleotides. However, we can compare these to YTH-"3A" point-mutant counterparts that disrupt m⁶A selectivity (Fig. 1).

We immunoprecipitated tagged YTH wild-type or "3A" mutant factors and performed qPCR for validated m⁶A targets or negative control transcripts. By comparison to control GFP-IP, we observed preferential binding of Ythdc1/Ythdf on multiple m⁶A targets, compared to non-m⁶A transcripts (Fig. 5B, C). By testing companion "3A" mutant factors, we gained evidence for direct association of YTH factors on m⁶A targets. However, a clear picture of target selectivity did not emerge (Fig. 5B, C). *Syx1A* and *aqz* exhibited the most clearly differential association between wt and 3A forms of both Ythdc1 and Ythdf. We observed potentially selective association with other loci, in that *gish* was preferentially bound only by wt Ythdc1 while *fl(2)d* was preferentially bound only by wt Ythdf.

We bear in mind these were ectopic experiments, and thus cannot rule out non-physiological associations. Even though we observed many cases of YTH-dependent target association, both YTH-3A proteins still exhibited apparent enrichment compared to GFP. If these mutant YTH proteins are still capable of incorporating into RNA granules, this may conceivably indicate indirect interactions with transcripts. Nevertheless, these data provide evidence that YTH domain proteins, including Ythdf, associate with specific m⁶A target transcripts via their m⁶A-binding pocket in *Drosophila* cells.

**m⁶A does not globally influence mRNA levels in *Drosophila*.** There is diverse literature on linking mammalian YTHDF homologs to RNA decay and/or translational activation, while the function of *Drosophila* m⁶A/Ythdf has been little studied. The only prior study integrated MeRIP-seq peaks from S2R+ cells with RNA-seq data from m⁶A pathway depletions, and concluded that m⁶A correlated with slightly elevated levels of target mRNAs at steady state[54]. With our high-stringency m⁶A map from heads, we generated RNA-seq data from one- and three-week old heads using *Mettl3*, *Ythdf* heterozygotes, and transheterozygotes. The heterozygote samples provide matched genetic backgrounds for comparison, and the temporal series assesses CNS stages including an advanced setting during which behavioral phenotypes were apparent (Figs. 2 and 3).

Transcriptome analyses revealed scores of differentially expressed genes in one- and three-week old mutants (Supplementary Dataset 5), a majority of which were uniquely misexpressed (Supplementary Fig. 10a, b). Most affected genes were not found to be common between m⁶A writer (*Mettl3*) and reader (*Ythdf*) mutants, although there were mild changes that gradually increased with tissue age (Supplementary Fig. 10c and d). Thus, there did not appear to be a clear signature of m⁶A/Ythdf regulation revealed by bulk gene expression.

We examined this more closely by directly examining the behavior of m⁶A targets. We reasoned that targets with systematically higher levels of methylation—that is, genes with increasing proportions of methylated transcripts—would be more sensitive to loss of the m⁶A pathway. However, while our miCLIP libraries provide Mettl3-dependent peaks and single nucleotide resolution mapping of m⁶A sites in the transcriptome, it is not possible to infer overall methylation levels. A solution to this limitation, grouping targets by number of sites/peaks, has been adopted by others[31,68] and proposes that targets with increasing numbers of peaks/sites may have more individual transcripts with at least one m⁶A modification. Therefore, we binned genes by numbers of Mettl3-dependent m⁶A peaks.

unaltered in *Mettl3[null]* samples (Fig. 5A′). These data provide stringent validation of our m⁶A maps.

Overall, our high-quality miCLIP data from the *Drosophila* head reveals that the position of m⁶A in this species appears distinct amongst metazoans (highly 5′ UTR specific) and occurs within a distinct adenosine-rich context.

***Drosophila* YTH factors associate with m⁶A targets in a YTH-dependent manner.** We next assessed association of m⁶A targets with YTH factors using transfected constructs in S2-S cells, a derivative of S2 cells lacking viruses[67]. Although overexpression may affect the localization properties of YTH domain proteins[31], we showed that ectopic Ythdc1 and Ythdf localize to the nucleus and cytoplasm of cultured cells, respectively[55]. In these tests, it is also relevant to consider that we are evaluating the association of

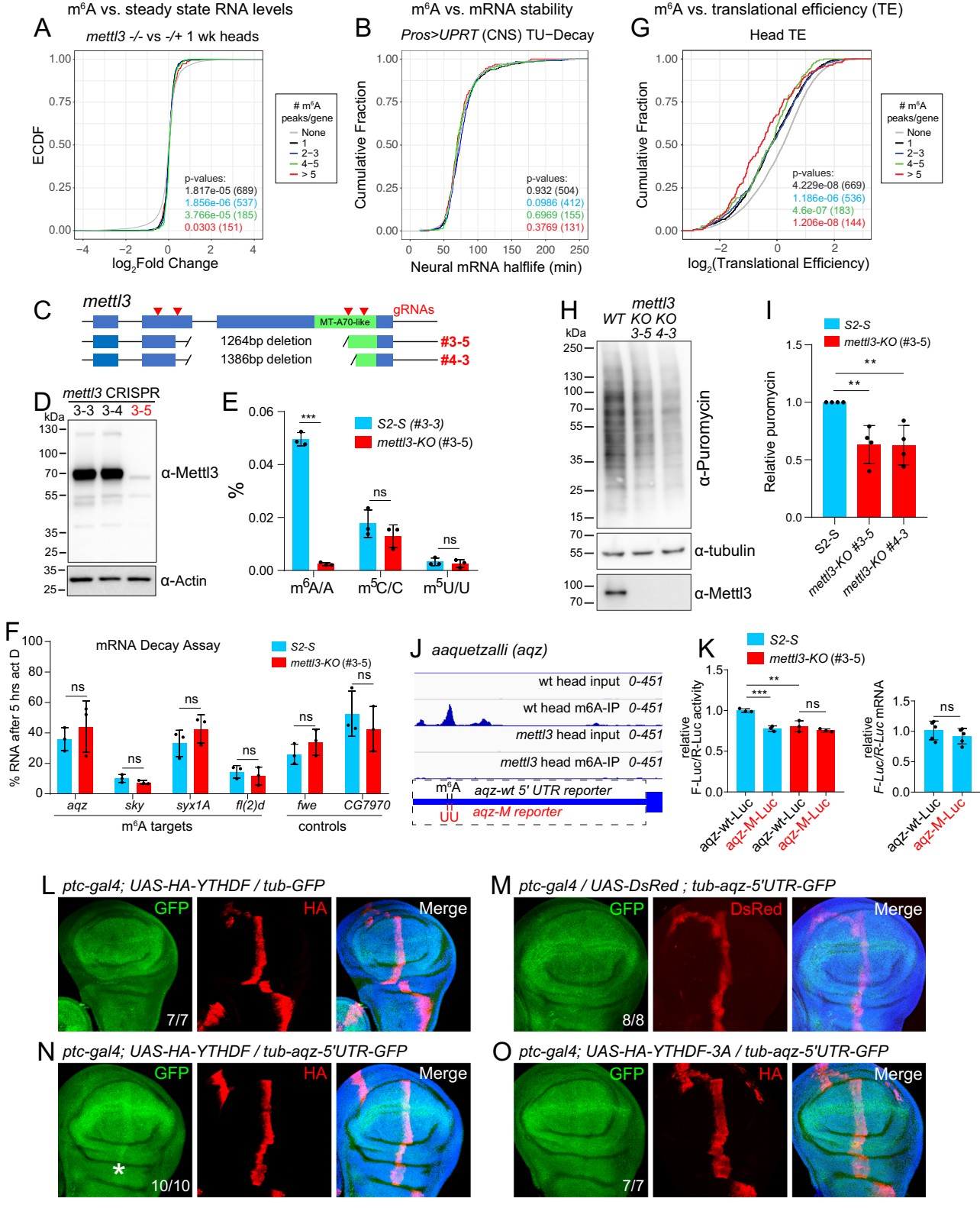

In contrast to prior association of *Drosophila* m⁶A with increased steady-state levels of targets[54], we did not observe many changes in our high-confidence m⁶A targets in *Mettl3* (Fig. 6A) or *Ythdf* mutant CNS from any stage (Supplementary Fig. 10e–h). Paradoxically, even though all bins of m⁶A targets clustered closely with a log₂ fold change of 0, Kolmogorov–Smirnov (KS) tests indicated statistical significance when comparing sets of m⁶A targets and background. While statistically different, our

analyses failed to detect any directional gene expression changes in methylated transcripts under writer or reader loss (Fig. 6A and Supplementary Fig. 10).

Since it was conceivable that some expression trends were masked in steady-state measurements, we examined a published dataset of in vivo mRNA decay rates generated using dynamic TU-tagging from the *Drosophila* CNS, obtained by pulse-chase labeling of *pros-Gal4 > UAS-UPRT* cells with 4-thiouridine[69]. We

**Fig. 6 m⁶A mediates translational activation in *Drosophila*.** **A** Steady-state RNA levels of m⁶A targets are not affected by loss of m⁶A. Differential gene expression analysis of 1-week-old *Mettl3* heterozygote and mutant heads. No directional change was observed in any group of m⁶A targets, binned by increasing numbers of Mettl3-dependent m⁶A peaks. **B** m⁶A targets are not biased in their mRNA stability. The half-life of neural genes obtained from TU-decay measurements[69] is plotted as cumulative distribution grouping target genes as in (**A**), based on number of peaks per target. **C** CRISPR/Cas9 mutagenesis of *Mettl3* in S2-S cells. **D** Western blotting shows lines #3–5 and #4–3 lack Mettl3 protein; #3–3 was used as a control line. **E** Quantitative liquid chromatography–mass spectrometry (LC–MS) and absolute quantification confirms specific lack of m⁶A in *Mettl3-KO* cells. Error bars, mean ± SD; $n = 3$ biological replicates, two-tailed *t*-test, ***$p < 0.001$. m⁶A $p = 5.87E{-}06$. n.s. = non-significant, m⁵C $p = 0.604$, m⁵U $p = 0.612$. **F** qPCR of m⁶A-modified mRNAs shows they have similar half-lives in *Mettl3-KO* and wild-type cells. Error bars, mean ± SD; $n = 3$ biological replicates, two-tailed *t*-test, ns, not significant. **G** Translation efficiency (TE) measurements[70] plotted as cumulative fractions for targets with different numbers of m⁶A peaks. m⁶A is preferentially deposited on genes with low TE. For panels **A**, **B**, and **G**, a bootstrap method generated the background distribution (None) using genes that lacked m⁶A peaks. To generate *p*-values, two-sided Kolmogorov–Smirnov (KS) tests were performed comparing the background distribution and each group of m⁶A target genes. Number of targets are included in parentheses. **H** Puromycin labeling (5′ in 5 µg/mL media, 50′ recovery) shows reduced global protein synthesis in *Mettl3-KO* cells. **I** Quantification of nascent protein synthesis in independent *Mettl3-KO* S2-S cells. Error bars, mean ± SD; $n = 4$ biological replicates, Two-tailed *t*-test, **$p < 0.01$. #3–5 $p = 0.00441$, #4–3 $p = 0.00496$. **J** *aqz*, a model m⁶A target. We assayed a wildtype *aqz* 5′ UTR reporter and a variant with two point mutations of m⁶A sites within the strongest Mettl3-dependent peak (*aqz-M*). **K** *aqz-wt-Luc* generates more reporter output than *aqz-M-Luc* in wildtype cells, but these have equivalent output in *Mettl3-KO* cells. Error bars, mean ± SD; $n = 3$ replicates (luciferase), $n = 5$ replicates (qPCR). Two-tailed *t*-test, **$p < 0.01$, ***$p < 0.001$. $p = 0.0006$, $p = 0.0089$. n.s. non-significant, F-Luc/R-Luc $p = 0.278$, mRNA F-Luc/R-Luc $p = 0.326$. This is one representative result from three independent repeats. **L–O** Wing imaginal disks expressing *tub-GFP* (in green), *ptc-Gal4 > UAS-HA-Ythdf* wild type or *3A* mutants or UAS-DsRed (HA/DsRed in red) and DAPI (blue). **L** Ectopic Ythdf has only marginal effects on the parental *tub-GFP* reporter. **M** DsRed does not affect the *tub-aqz-5′UTR-GFP* reporter. **N–O** Wildtype Ythdf (**N**) but not Ythdf-3A (**O**) enhances GFP production from *tub-aqz-5′UTR-GFP*. Numbers of disks analyzed are labeled, and representative results are shown. Scale bar is 100 µM. Source data are provided as a Source Data file.

observed that, in aggregate, m⁶A-modified transcripts had identical mRNA half-lives as the background distribution (Fig. 6B). Thus, we were not able to discern global m⁶A regulatory impacts on transcript properties.

To test this further, we used CRISPR/Cas9 to delete the *Mettl3* locus from S2-S cells. Western blotting validated absence of Mettl3 protein in multiple independent cell lines (Supplementary Fig. 11), including two lines that we used for further analysis (#3-5 and #4-3, Fig. 6C, D). Because #4–3 retained genomic material internal to the confirmed deletion, despite absence of Mettl3 protein (Supplementary Fig. 11), we generally relied on line #3-5 (*Mettl3-KO*). Moreover, we directly measured $N^6$-m⁶A levels in *Mettl3-KO* cells using quantitative liquid chromatography–mass spectrometry (LC–MS). External calibration curves prepared with A and m⁶A standards determined the absolute quantities of each ribonucleoside. The mRNA m⁶A methylation levels in knockout cells were <5% of those in wild-type cells, whereas other modified ribonucleosides were unaffected (Fig. 6E).

Using *Mettl3-KO* cells, we performed RNA decay assays of validated m⁶A targets and control transcripts. Following inhibition of transcription using actinomycin D, we observed a range of transcript levels across different loci, but none of these were significantly different between wild-type and m⁶A-deficient cells (Fig. 6F). Overall, our analyses using S2 cells and intact nervous system indicate that mRNA stability of m⁶A-containing transcripts is neither substantially nor directionally influenced by loss of m⁶A in *Drosophila*, in contrast to m⁶A in mammals.

**m⁶A is preferentially deposited on fly transcripts with lower translational efficiency.** In light of these data, we examined the alternate possibility of m⁶A-dependent translational control. For this purpose, we utilized ribosome-profiling datasets from *Drosophila* heads[70] to assess translational efficiencies of transcripts with or without m⁶A modifications. Strikingly, we found that genes with m⁶A had lower translational efficiency than the background distribution (Fig. 6G). The functional relevance of this observation was strengthened by the fact that the number of m⁶A peaks per transcript exhibited a progressive, inverse correlation with translational efficiency and contrasted with the lack of correlation of m⁶A modification with either steady-state transcript levels or RNA stability. Altogether, these results suggest

that m⁶A mediates translational control. Moreover, as our miCLIP maps were generated from highly dT-selected RNAs, we infer that this may reflect modifications that are mostly present in cytoplasmic transcripts available for binding to Ythdf.

***Drosophila* m⁶A enhances target protein output.** The dominant location of *Drosophila* m⁶A in 5′ UTRs, contrasting with the preferred residence of mammalian m⁶A in 3′ UTRs, is suggestive of a role in influencing translation. However, the above genomic analyses are correlational in nature, and do not directly connect m⁶A to gene regulation. One scenario is that m⁶A, being enriched amongst poorly translated mRNAs, is a suppressive mark. However, an alternative logic is that m⁶A is a positive mark that is preferentially deposited on transcripts that are inefficiently translated, which might make the potential impact of enhancement more overt. Such logic was proposed for mammalian YTHDC2 to enhance translation of low efficiency translated m⁶A targets[38].

We first evaluated if m⁶A might exert global impact on translation. We exploited our *Mettl3-KO* S2-S cells and monitored newly synthesized proteins using puromycin incorporation[71]. Interestingly, we observe a difference between steady-state protein accumulation and nascent protein synthesis in wild-type and m⁶A-mutant cells. In particular, when analyzing similar amounts of total cellular protein, we observed that both *Mettl3-KO* cells (#3-5 and #4-3) consistently generated less newly synthesized bulk proteins than did control S2-S cells (Fig. 6H, I). Western blotting for tubulin, a non-m⁶A target, verified similar steady-state accumulation between wild-type and knockout cells, while Mettl3 blotting confirmed knockout cell status. These data suggested that m⁶A may enhance protein output.

We tested this further using a reporter assay. One of the most prominent m⁶A targets was *aaquetzalli* (*aqz*), whose 5′ UTR bore highly *Mettl3*-dependent miCLIP peaks with multiple CIMS (Fig. 6J), and whose modification we had validated (Fig. 5). Aqz is required for cell polarity and neural development[72]. We cloned its 5′ UTR upstream of firefly luciferase (*aqz-wt-Luc*), as well as a companion mutant version in which we mutated both identified m⁶A sites within the strongest miCLIP peak (*aqz-M-Luc*). We co-transfected these reporters with renilla control reporter into S2-S cells, and observed that the wildtype *aqz* reporter reliably yielded

higher output (Fig. 6K). However, when we repeated these tests in *Mettl3-KO* cells, the wildtype and mutant *aqz* reporters were indistinguishable (Fig. 6K). Finally, we tested the transcript levels of the reporters by qPCR. While these measurements were more variable than the luciferase activity readouts, they were not significantly different between wt and mutant *aqz* reporters (Fig. 6K). These tests provide evidence that individual 5′ UTR m⁶A sites can confer activation in fly cells, and are consistent with translational regulation.

**Drosophila Ythdf enhances output of an m⁶A reporter in a YTH-dependent manner.** To test if Ythdf might be an effector of m⁶A-mediated target activation, we implemented a transgenic assay. We used a reporter backbone consisting of GFP under control of the tubulin promoter (*tub-GFP*), a transgene that is broadly and relatively evenly expressed in the animal[73]. In this genetic background, we can coexpress factors in a spatially defined subpattern, to assess regulatory impact on the transgene. When we stain wing imaginal disks bearing a naive reporter, and expressing UAS-HA-Ythdf along the anterior–posterior boundary (using *ptc-Gal4*), we do not observe substantially different GFP protein accumulation in cells co-expressing Ythdf, compared to non-Gal4 cells as internal control territories (Fig. 6L).

Since we had validated *aqz* as both an m⁶A target (Fig. 6K) and a Ythdf target (Fig. 5), we transferred its 5′ UTR into the *tub-GFP* reporter. The *tub-aqz-5′UTR-GFP* transgene expressed GFP broadly and the levels were not noticeably different from the parent transgene. Its accumulation was not affected by co-expression of a UAS-DsRed transgene (Fig. 6M). However, when we introduced into the *ptc > HA-Ythdf* background, GFP was elevated specifically within the Ythdf-expressing domain (Fig. 6N). This was consistent with a role for Ythdf in enhancement of this m⁶A target.

To test if this was due to specific activity of Ythdf, we generated an HA-tagged transgene containing the three YTH pocket mutations, which we showed abrogates association to m⁶A in vitro (Fig. 1C) and to validate m⁶A-bearing transcripts in cells (Fig. 5C). HA-Ythdf-3A protein accumulated to a similar level as wild type, and was also similarly neutral as its wild-type counterpart when tested on the parent *tub-GFP* reporter. Moreover, mutant Ythdf-3A was unable to enhance GFP protein output from the *tub-aqz-5′UTR-GFP* transgenic reporter (Fig. 6O). These data support the notion that Ythdf recognizes m⁶A-bearing 5′UTR targets for translational enhancement.

**An autonomous, m⁶A-dependent, neural function for Ythdf in memory.** The availability of wild-type and mutant Ythdf transgenes allowed us to conduct further genetic tests of the connection between m⁶A readout and STM. Overexpression of either Ythdf transgene, (e.g. with *ptc-Gal4, ap-Gal4*) did not substantially impair viability or developmental patterning; and pan-neuronal expression of Ythdf using *elav-Gal4* did not affect lifespan (Supplementary Fig. 4). Therefore, even though we could detect a selective m⁶A-dependent impact of Ythdf on reporter transgenes, elevated Ythdf expression does not interfere with normal developmental programs, or has effects that are otherwise within the range of developmental compensation. This mirrors the lack of substantial consequences of removing Ythdf.

We moved to perform cell-type-specific transgenic rescue assays. Building on our observation that neural-knockdown of Ythdf phenocopied the STM defects seen in whole-animal mutations (Fig. 3G), we introduced *elav-Gal4* and *UAS-Ythdf-wt* or *UAS-Ythdf-3A* transgenes into *Ythdf* hemizygous null

backgrounds. The *Ythdf* mutants carrying *elav-Gal4* had defective STM, indicating the Gal4 transgene does not improve this behavioral output (Fig. 7A). This was important to rule out, since at least some other *Drosophila* neuronal phenotypes are modified by Gal4 alone[74,75]. With this control background as reference, we found that the STM deficit of *Ythdf* nulls could be rescued by pan-neuronal restoration of Ythdf, thereby restoring normal STM (Fig. 7A). In contrast, *elav > Ythdf-3A* transgenes did not rescue normal STM capacity to *Ythdf* mutants (Fig. 7A). Thus, m⁶A binding is critical for neural Ythdf function during STM formation.

**An m⁶A/Ythdf pathway functions specifically in MB to promote memory.** In *Drosophila*, associative olfactory memory is stored in the intrinsic neurons of the MB[9,76]. Bearing in mind that Ythdf-mediated regulation has particular impact on STM, we evaluated the effects of its gain-of-function on wild-type MB neurons using *MB247-Gal4*. Strikingly, *MB247-Gal4 > Ythdf* flies exhibited compromised STM formation at 20 days, while *MB247-Gal4 > Ythdf-3A* flies were normal (Fig. 7B). This behavioral defect was relatively specific: flies overexpressing wild-type and mutant Ythdf exhibited similar locomotor activity when quantified over 60 s of tracking (Fig. 7C). Thus, ectopic Ythdf disrupts STM in an m⁶A-dependent manner.

These data prompted us to investigate the endogenous m⁶A/Ythdf pathway with respect to the MB more rigorously. Flies depleted of Mettl3 in the MB exhibited STM impairment (Fig. 7D), comparable to whole-animal *Mettl3* null mutants (Fig. 2). Similarly, *MB247-Gal4 > UAS-Ythdf[RNAi]* flies exhibited substantial STM defects (Fig. 7E), comparable to *Ythdf* null animals (Fig. 3F, G). Thus, Mettl3 and Ythdf are specifically required in MB neurons to facilitate normal STM.

It was possible that these m⁶A factors act not only in MB neurons, but also in other neural populations, to promote STM. To address the sufficiency of the m⁶A/Ythdf pathway in MB neurons, we conducted further transgenic rescue tests. Recall that we were able to rescue *Mettl3* null animals (Fig. 2G) and *Ythdf* null animals (Fig. 7A) by pan-neuronal expression of these factors. Now, we tested whether we could do so by expressing them only using *MB247-Gal4*. Indeed, this fully rescued the STM defects of *Mettl3* mutants (Fig. 7F) and substantially improved those of *Ythdf* mutants (Fig. 7G). Therefore, even though m⁶A regulation undoubtedly impacts most cell types in the animal, the m⁶A/Ythdf pathway plays a cell-autonomous role in MB neurons to mediate STM.

To investigate whether this might be potentially associated with effects on MB structure, we introduced a *UAS-GFP* transgene into the *MB247-Gal4, Mettl3[null]* mutant background, which facilitated visualization of the mushroom-body lobes. In *Mettl3* heterozyotes, the MB adopts its characteristic morphology: the left and right MB horizontal lobes are distinct (Fig. 7H, top left). However, a majority of *Mettl3* mutants exhibit fusion of the MB β-lobes (Fig. 7H, top right). We did not observe comparable fusion of the γ-lobes or shortening of the α-lobes, indicating a spatially restricted effect. We initially conducted these tests at 3 weeks, to align with our functional tests of learning and memory. However, preferential fusion of *Mettl3* mutant MB β-lobes was also detected at 1 week, even though their behavioral performance was normal at this time (Supplementary Fig. 3). Accordingly, we tested *Ythdf* heterozygous and hemizygous mutants, and similarly observed qualitatively similar fusion of MB β-lobes in mutants (Fig. 7I), at both 1 and 3 weeks. Our MB staining data of the two mutants at both timepoints are quantified in Fig. 7J.

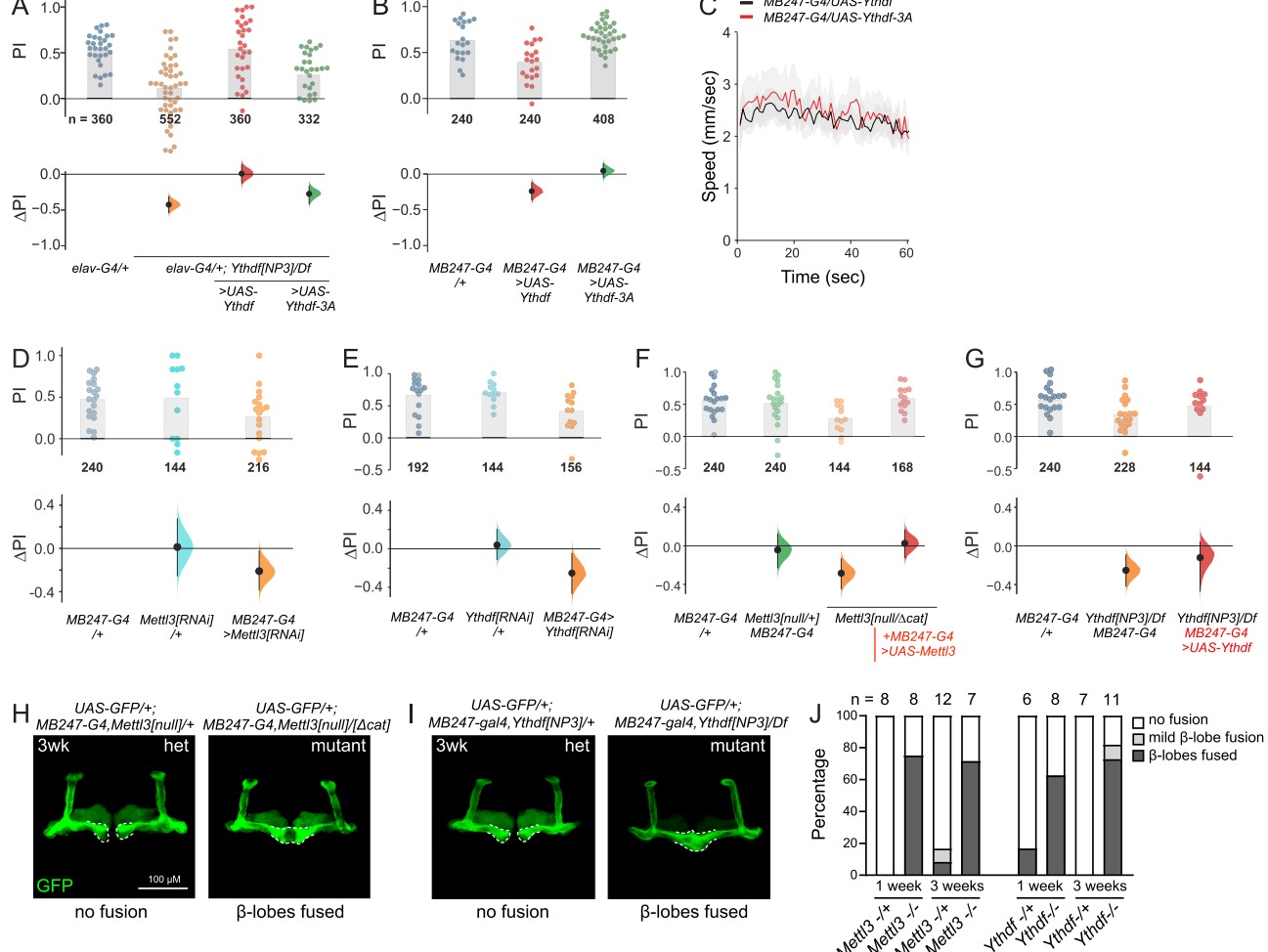

**Fig. 7 An m⁶A/Ythdf pathway acts in the mushroom body to mediate STM. A–C** In vivo function of Ythdf during learning and memory requires m⁶A-binding capacity. **A** Pan-neuronal expression (using *elav-Gal4, =G4*) of wild-type Ythdf, but not Ythdf-3A, rescued STM defects in *Ythdf* hemizygote mutants at 20 days. *Elav-G4; Ythdf[NP3]/Df* vs. *elav-G4/+* = −0.42[95CI −0.53, −0.31] $p < 1*10^{-4}$. *Elav-G4; UAS-Df1; Ythdf[NP3]/Df* vs. *elav-G4/+* = 0 [95CI −0.13, +0.14] $p = 0.684$. *Elav-G4; UAS-Df1-3A; Ythdf[NP3]/Df* vs *elav-G4/+* = −0.27[95CI −0.41, −0.17] $p < 1*10^{-4}$. **B** Mushroom body (MB) expression (using *MB247-Gal4*) of Ythdf, but not Ythdf-3A, impairs STM in 20-day flies. *MB247-G4; UAS-Ythdf* −0.24[95CI −0.37, −0.12] $p = 0.001$. *MB247-G4/+* vs. *MB247-G4; UAS-Ythdf-3A* (vs. *MB247-G4/+* = +0.04[95CI −0.05, +0.15] $p = 0.415$. **C** Expression of Ythdf-wt/3A proteins does not affect locomotion behavior. **D** Depletion of *Mettl3* in the MB phenocopied STM impairment seen in whole animal *Mettl3* mutants (see Fig. 2). *Mettl3* [RNAi]/+ vs. *MB247-G4/+* = +0.01[95CI −0.26, +0.27] $p = 0.654$. *MB247-G4; UAS-Mettl3[RNAi]* vs. *MB247-G4/+* = −0.21[95CI −0.38, −0.02] $p = 0.034$. **E** MB-specific knockdown of *Ythdf* compromised STM, similar to what has been observed in whole animal *Ythdf* mutants (see Fig. 3). *Ythdf[RNAi]/+* vs. *MB247-G4/+* = +0.04[95CI −0.11, +0.2] $p = 0.835$. *MB247-G4; UAS-Ythdf[RNAi]* vs *MB247-G4/ +* −0.25[95CI −0.46, −0.05] $p = 0.002$. **F** Re-expression of *Mettl3* only in MB neurons rescued STM defects in *Mettl3* trans-heterozygote mutants. *MB247-G4; UAS-Mettl3[null]/+* vs. *MB247-G4/+* = −0.04[95CI −0.23, +0.12] $p = 0.86$. *Mettl3[null]/Mettl3[Δcat]* vs. *MB247-G4/+* = −0.28[95CI −0.45, −0.13] $p = 0.004$. *MB247-G4/UAS-Mettl3; Mettl3[null]/Mettl3[Δcat]* vs. *MB247-G4/+* = +0.02[95CI −0.12, +0.17] $p = 0.766$. **G** Re-expression of *Ythdf* only in MB neurons improves STM defects in *Ythdf* hemizygous mutants. *MB247-G4; Ythdf[NP3]/Df* vs. *MB247-G4/+* = −0.25[95CI −0.41, −0.09] $p = 0.005$. *MB247-G4/UAS-Ythdf; Ythdf[NP3]/Df* vs. *MB247-G4/+* = −0.12[95CI −0.47, +0.03] $p = 0.572$. STM and locomotion assays were conducted in 20-day-old flies. Each dot in the STM scatter plots represents a PI average of 12 flies. All control–test differences are displayed as effect sizes with error curves and 95% confidence intervals. No null-hypothesis significance testing was performed; two-tailed Mann–Whitney *P* values are shown for legacy purposes only. For multiple comparisons, several test groups were compared against a common control group. **H** and **I** MB structural defects in *Mettl3* (**H**) and *Ythdf* (**I**) mutants at 3 weeks. All panels depict GFP driven by *MB247-Gal4*. **H** Representative *Mettl3* heterozygote (left) and trans-heterozygote mutant (right), showing fusion of β-lobes in the latter. **I** Representative *Ythdf* heterozygote (left) and hemizygote mutant (right), showing fusion of β-lobes in the latter. **J** Quantification of genotypes analyzed in **H** and **I** at 1 and 3 weeks; *n* sizes are labeled. Source data are provided as a Source Data file.

At this point, it remains to be determined whether the MB structural defects are diagnostic and a more sensitive readout that precedes subsequent behavioral deficits, or whether these phenotypes occur in parallel. In either scenario, these tests provide substantial evidence that a *Drosophila* m⁶A/Ythdf pathway maintains MB structure, and operates cell-autonomously in the MB to mediate odor-avoidance learning.

## Discussion

**Distinct local contexts, genic location, and regulatory impact for m⁶A in different metazoans.** Despite tremendous interests in the regulatory utilities and biological impacts of mRNA methylation, there has been relatively little study from invertebrate models. Given that the m⁶A pathway seems to have been lost from *C. elegans*, *Drosophila* is an ideal choice for this. Since the

initial report that *Mettl3* mutants affect germline development[52], we and others showed that *Drosophila* harbors an m⁶A pathway similar to that of mammals, but simplified in that it has a single nuclear and cytoplasmic YTH reader[53–55]. Nevertheless, *Drosophila* has proven to be a useful system to discover and characterize novel m⁶A factors[54,55,77,78]. Expanding the breadth of model systems can increase our appreciation for the utilization and impact of this regulatory modification.

It is widely presumed, based on mammalian profiling, that metazoan m⁶A is enriched at stop codons and 3′ UTRs. However, our high-resolution maps indicate that 5′ UTRs are by far the dominant location of methylation in mature *Drosophila* mRNAs. Although further study is required, many of these m⁶A 5′ UTR regions coincide with our previous embryo miCLIP data (e.g. Supplementary Fig. 7), while other miCLIP CIMs calls located in other transcript regions[55] proved usually not to be Mettl3-dependent. Thus, our data indicate a fundamentally different distribution of m⁶A in *Drosophila* mRNAs compared to mammals.

While mammalian m⁶A clearly elicits a diversity of regulatory consequences, depending on genic and cellular context and other factors, a dominant role is to induce target decay through one or more cytoplasmic YTH readers. This harkens back to classic observations that m⁶A is correlated with preferential transcript decay[79], and more recent data that loss of m⁶A writers[80–82] or cytoplasmic YTH readers[29,32] results in directional upregulation of m⁶A targets. However, several lines of study did not yield convincing evidence for a broad role for the *Drosophila* m⁶A pathway in target decay. Instead, the dominant localization of m⁶A in fly 5′ UTRs is suggestive of a possible impact in translational regulation. Our genomic and genetic evidence support the notion that m⁶A is preferentially deposited in transcripts with overall lower translational efficiency, but that m⁶A/Ythdf may potentiate translation. However, we can rationalize a regulatory basis for these apparently opposite trends, if the greater modulatory window of poorly translated loci is utilized for preferred targeting by m⁶A/Ythdf.

As is generally the case for mammalian m⁶A, the choice of how appropriate targets are selected for modification, and which gene regions are preferentially methylated, remains to be understood. The minimal context for m⁶A is insufficient to explain targeting, and as mentioned also seems to be different between *Drosophila* and vertebrates. A further challenge for the future will be to elucidate a mechanism for m⁶A/Ythdf-mediated translational regulation. This will reveal possible similarities or distinctions with the multiple strategies proposed for translational regulation by mammalian m⁶A, which include both cap-independent translation via 5′ UTRs during the heat-shock response via eIF3[83] or YTHDF2[36]; cap-dependent mRNA circularization via Mettl3-eIF3H[84]; and activity-dependent translational activation in neurons[34].

**Roles for the m⁶A/DF1 pathway in learning and memory**. Recent studies have highlighted neuronal functions of mammalian m⁶A pathway factors[39]. There is a growing appreciation that mouse mutants of multiple components in the m⁶A RNA-modification machinery affect learning and memory[34,46,48–50]. Here, we provide substantial evidence that, in *Drosophila*, neural m⁶A is critical for STM. We specifically focused on STM as this paradigm has been extensively characterized in *Drosophila*. Mouse studies have almost exclusively examined effects on LTM, and these two memory phases are mechanistically distinct[85,86]. One main distinction is that LTM requires protein synthesis after training, while STM does not. So, while direct comparisons between the two systems are not possible, it is

nevertheless instructive to consider the parallels and distinctions of how m⁶A facilitates normal memory function in these species. This is especially relevant given that both mouse and fly central nervous systems require a cytoplasmic YTH factor for memory.

In mice, the m⁶A writer Mettl3 enhance long-term memory consolidation, potentially by promoting the expression of genes such as *Arc*, *c-Fos* and others[46]. Another study found that Mettl14 is required for LTM formation and neuronal excitability[49]. Conversely, knockdown of the m⁶A demethylase FTO in the mouse prefrontal cortex resulted in enhanced memory consolidation[48]. Amongst mammalian YTH m⁶A readers, YTHDF1 was shown to induce the translation of m⁶A-marked mRNA specifically in stimulated neurons[34]. In cultured hippocampal neurons, levels of YTHDF1 in the PSD fraction were found to increase by ~30% following KCl treatment. This suggests that YTHDF1 concentration at the synapse could be critical for regulating the expression levels of proteins (such as CaMK2a) involved in synaptic plasticity[87]. Taken together, these studies suggest the m⁶A pathway is a crucial mechanism of LTM consolidation in mammals that optimizes animal behavioral responses.

Of note, the genetics and sample sizes possible in *Drosophila* permit comprehensive, stringent, and anatomically resolved analyses[88]. Thus, in our study, we systematically analyze all writer and reader factors, and reveal a notable functional segregation, suggesting that the cytoplasmic reader Ythdf is a major effector of Mettl3/Mettl14 m⁶A in memory. Given that *Ythdf* mutants otherwise exhibit few overt developmental or behavioral defects in normal or sensitized backgrounds (while *Ythdc1* mutants generally phenocopy *Mettl3/Mettl14* mutants) its role in STM is a surprising insight into the contribution of Ythdf to a critical adaptive function. Moreover, we can pinpoint the spatial requirements of m⁶A for STM, by showing that (1) neuronal-specific and MB-specific depletion of *Mettl3/Ythdf* can induce defective STM, and (2) neuronal and MB-specific restoration of Mettl3 or Ythdf to their respective whole-animal knockouts restores normal STM. Moreover, the fact that Ythdf gain-of-function in the MB can also disrupt STM, but does not generally alter other aspects of development or behavior, points to a homeostatic role of m⁶A regulation in *Drosophila* learning and memory.

We observed that STM defects in fly m⁶A mutants are age-dependent, which has not been reported in mammals. Although many physiological capacities decline with life history, the observed STM defects seem to be decoupled from other age-related phenotypes, since mutation of *Ythdf* or neural over-expression of Ythdf can interfere with STM but does not substantially impact lifespan or locomotion. In this regard, *Mettl3 and Ythdf* are different from classical memory genes such as *rutabaga*[89] because STM impairment in m⁶A mutants was absent in young flies and only became apparent with progressing age.

One interpretation is that there is a cumulative effect of deregulated m⁶A networks that has a progressive impact specific to mushroom-body neurons. To gain further mechanistic insights, future studies will need to examine age-related changes in gene expression and/or translation, in a cell-specific manner. It remains to be seen whether specific deregulated targets downstream of Ythdf have large individual effects, or whether the STM deficits arise from myriad small effects on translation. Ythdf-CLIP and ribosome profiling from the CNS may prove useful to decipher this. Assuming that loss of translational enhancement of m⁶A/Ythdf targets mediates STM defects, one possibility, to be explored in future studies, is that some targets may already be known from prior genetic studies of memory[5].

## Methods

**m⁶A reader constructs**. We obtained full-length cDNAs obtained by PCR from a cDNA library for Ythdc1 (encoding Ythdc1-PA, 721aa), and Ythdf (encoding Ythdf-PA, 700aa) into pENTR vector. We then used site-directed mutagenesis (primers listed in Supplementary Dataset 6) to generate pENTR-Ythdc1-3A (w276A w327A L338A), and pENTR-Ythdf-3A (w404A-W459A-w464A). These were transferred into the *Drosophila* Gateway vector pAGW (N-terminal GFP fusion) and pAHW (N-terminal HA tag) to make all combinations of tagged wild-type and "3A" mutant versions for expression in *Drosophila* cell culture. We also cloned the Ythdf sequences into pTHW (N-terminal HA fusion) to generate UAS-HA-Ythdf and UAS-HA-Ythdf-3A for transgenes. For mammalian expression, we cloned wild-type and "3A" mutant versions of Ythdc1 and Ythdf into pcDNA5/FRT/TO with an N-terminal 3xFlag-tag.

**m⁶A probe pulldown with fly Ythdf and Ythdc1**. As we were unable to purify sufficient amounts of full-length *Drosophila* Ythdf and Ythdc1 proteins from *Drosophila* cell culture for in vitro characterization, we instead turned to purifying them from mammalian cells. For initial interaction tests of wild-type Flag-Ythdc1/DF with A/m⁶A RNA probes, we seeded 4 million HEK293T cells in a 10 cm dish 24 h prior to calcium phosphate transfection, and used 10 μg of plasmid DNA (pcDNA-3xFlag-Ythdc1/DF constructs) per 10 cm dish. Cells were harvested 24 h post transfection. Cells from one 10 cm dish and 0.6 mL of lysis buffer were used per condition. Cells were lysed in NP-40 lysis buffer (50 mM Tris–HCl pH 7.5, 150 mM NaCl, 0.5% NP-40, 5 mM MgCl₂, EDTA-free protease inhibitor tablet (Roche), 1 mM PMSF) on ice. Clarified lysate was then incubated with A or m⁶A containing RNA probe (Supplementary Dataset 6) on ice at 1 μM concentration for 20 min. Reactions were then irradiated with 365 nm UV (Spectroline ML-3500S) on ice for 10 min. The reaction was then incubated with 60 μL of high capacity streptavidin agarose 50% bead slurry (Pierce #20357) at 4 °C on a rotatory wheel for 3 h. The beads were then washed with 1% SDS in TBS (3 × 1 mL), 6 M urea in TBS (3 × 1 mL), and TBS (3 × 1 mL). The RNA-bound proteins were eluted by boiling the beads in 50 μL of 1× Laemmli sample buffer (80 mM Tris–HCl pH 6.8, 2% SDS, 10% glycerol, 5% B-mercaptoethanol, 0.02% bromophenol blue) at 95 °C for 5 min. The input, flow-through, and eluates were separated by SDS–PAGE and analyzed by Western blotting using mouse α-Flag M2 (Sigma #F1804).

In the experiment where 3A mutants of Ythdf and Ythdc1 were tested, cells from two 10 cm dishes and 1.2 mL of lysis buffer were used per condition. The cross-linking and pull-down were performed as described above. AAACU and AA-m⁶A-CU sequences were used for this experiment.

### Proteomic profiling of the fly m⁶A interactome

*Mass spectrometry analysis.* For proteomics experiments with *Drosophila* S2 cells, we adapted our previously described method[57]. S2 cells were lysed by cryomilling. The resulting cell powder (750 mg) was first extracted with 1.5 mL of low-salt extraction buffer (20 mM Tris–HCl pH 7.5, 10 mM NaCl, 2 mM MgCl₂, 0.5% Triton X-100, 10% glycerol, protease inhibitor tablet (Roche), and phosphatase inhibitor (Pierce)), and then 1 mL of high-salt extraction buffer (50 mM Tris–HCl pH 7.5, 420 mM NaCl, 2 mM MgCl₂, 0.5 % Triton X-100, 10% glycerol, protease inhibitor tablet (Sigma) and phosphatase inhibitor (Pierce)). Low-salt and high-salt extracts were pooled, and protein concentration was determined by Bradford assay. The pooled extract was diluted to 3 mg/mL if needed before proceeding to photo-crosslinking.

AAACU or AA-m⁶A-CU oligo probe was added to 2 mL of extract to a final concentration of 1 μM. The reactions were incubated on ice for 20 min prior to photo-cross-linking. The reactions were then irradiated with 365 nm UV (Spectroline ML-3500S) on ice for 15 min. The reaction was then incubated with 60 μL of high capacity streptavidin agarose 50% bead slurry (Pierce #20357) at 4 °C on a rotatory wheel for 3 h. The beads were then washed with 1% SDS in TBS (3 × 1 mL), 6 M urea in TBS (3 × 1 mL), and TBS (3 × 1 mL). The RNA-bound proteins were eluted with RNase cocktail (Thermo Fisher) in RNase elution buffer (10 mM Tris–HCl pH 7.5, 40 mM NaCl, 1 mM MgCl₂) at 37 °C for 30 min with periodic agitation.

The proteomics files were searched against *Drosophila melanogaster* database downloaded from UniProt (https://www.uniprot.org/). To plot the mass spectrometry data (Fig. 1d), we first removed 242 proteins that were not consistently recovered in both replicate datasets, leaving 353 proteins. To calculate enrichment ratios for proteins identified by only one probe, we added 1 to all spectral count values. Proteins that do not exhibit differential binding to the A/m⁶A probes cluster around the plot origin, and we thresholded at 2.0× the interquartile range.

### Analysis of m⁶A by liquid chromatography-coupled mass spectrometry.

Total RNA was extracted from *Drosophila* S2 cells and whole female fly (one-week old) using TRIzol reagent and subjected to DNase treatment (Thermo Fisher #AM1907). The mRNA was then isolated through two rounds of poly-A selection using the oligo-d(T)₂₅ beads (NEB #S1419S). The RNA was digested with nuclease P1 (Wako USA #145-08221) and dephosphorylated with Antarctic phosphatase (NEB #M0289S). Briefly, 1 μg of RNA was digested with 2 units of nuclease P1 in buffer containing 7 mM NaOAc pH 5.2, 0.4 mM ZnCl₂ in a total volume of 30 μL at 37 °C for 2 h. 3.5 μL of 10× Antarctic phosphatase buffer and 1.5 μL of Antarctic

phosphatase was then directly added to the reaction and incubated at 37 °C for another 2 h.

Quantitative LC–MS analysis of m⁶A was performed on an Agilent 1260 Infinity II HPLC coupled to an Agilent 6470 triple quadrupole mass spectrometer in positive ion mode using dynamic multiple reaction monitoring (DMRM). The ribonucleosides in the digested RNA samples were separated by a Hypersil GOLD™ C18 Selectivity HPLC Column (Thermo Fisher #25003-152130; 3 μm particle size, 175 Å pore size, 2.1 × 150 mm; 36 °C) at 0.4 mL/min using a solvent system consisting of 0.1% formic acid in H₂O (A) and acetonitrile (B) based upon literature precedent[90]. The operating parameters for the mass spectrometer were as follows: gas temperature 325 °C; gas flow 12 L/min; nebulizer 20 psi and capillary voltage 2500 V, with fragmentor voltage and collision energy optimized for each different nucleoside. The nucleosides were identified based on the transition of the parent ion to the deglycosylated base ion: m/z 282 → 150 for m⁶A and m/z 268 → 136 for A. Calibration curves were constructed for each nucleoside using standards prepared from commercially available ribonucleosides. The level of m⁶A was determined by normalizing m⁶A concentration to A concentration in the sample.

***Drosophila* stocks**. *Ythdf[NP3]*, *FRT40A* and *Mettl14[sk1]/Tb-RFP cyo,w+* were previously described[55]. *Ythdc1[ΔN]/Dfd-YFP*, *TM3*; *Mettl3[Δcat]/TM6C*, *Mettl3 [null]/TM6C*, *Mettl14[fs]/Tb-RFP cyo,w+*; *UAS-Mettl3-HA* were a gift of Jean-Yves Roignant[54]. All of these were genotyped in trans to deficiencies (see Supplementary Dataset 6) to confirm the absence of the wild-type allele. Other stocks were obtained from the Bloomington *Drosophila* Stock Center. Deficiency lines: *Df(3R) Exel6197* (BL-41590, removes *Mettl3*), *Df(3L)ED208* (BL-34627, removes *Ythdc1*), *Df(3R)BSC461* (BL-24965, removes *Ythdf*) and *Df(3R)BSC655* (BL-26507, removes *Ythdf*). TRiP knockdown lines: *Mettl3* (BL-41590), *Mettl14* (BL-64547), *Ythdc1* (BL-34627) and *Ythdf* (BL-55151). Gal4 lines: *elav-gal4* (BL-8765), *elav[C155]-Gal4* (BL-458), *ptc-Gal4* (BL-2017), *tub-Gal4* (BL-5138), *da-Gal4* (BL-55851), *ap-Gal4* (BL-3041), *MB247-Gal4* (mef2-Gal4, BL-50742).

To generate the *tub-aqz-GFP* reporter, we cloned the *aqz* 5′UTR (chr3R:8,818,731-8,820,682) into the 5′UTR position of the tub-GFP vector (KpnI/BamHI). tub-aqz-GFP, UAS-HA-Ythdf and UAS-HA-Ythdf-3A (described above) were injected into *w[1118]* with Δ2-3 helper plasmid to obtain transformants (Bestgene, Inc.)

Flies were raised on standard cornmeal-based food medium containing 1.25% w/v agar, 10.5% w/v dextrose, 10.5% w/v maize, and 2.1% w/v yeast at 60% relative humidity.

**Survival experiments**. All *Drosophila* survival experiments were performed with mated female flies at 23 °C. Throughout the lifespan assessment, flies were kept in vials in groups of 10 and transferred to a new food vial every second or third day. The number of surviving flies was counted after each transfer. Average lifespan was calculated using the DABEST estimation statistics package[91]. The data were plotted to compare the average survival of each tested genotype against the average survival of the *w[1118]* control stock that was assayed in parallel.

**The MOT**. The MOT apparatus was designed to allow the monitoring of *Drosophila* behavior throughout olfactory conditioning in a controlled environment[16]. Flies were assayed in conditioning chambers, whereby the arena of each chamber was 50 mm long, 5 mm wide, and 1.3 mm high (Supplementary Fig. 2a). The floor and ceiling of each chamber was composed of a glass slide printed with transparent indium tin oxide electrodes (Walthy, China). Each side of the electrode board was sealed by a gasketed lid that formed a seal around the gap between the electrode board and the chamber wall. Facilitated by carrier air, the odors entered the chamber via two entry pipes and left the chamber through two vents that were located in the middle of the chamber. Up to four MOT chambers were stacked onto a rack which was connected to the odor and electric shock supply (Supplementary Fig. 2b). Chambers were illuminated from the back by two grids of infrared LEDs. Behavior inside the chambers was recorded with an AVT F-080 Guppy camera (Allied Vision) that was connected to a video acquisition board (PCI-1409, National Instruments). Electric shock during odor presentation is delivered when the animals walk on the electrode contacts. Olfactory preference was measured by tracking the movement of individual flies and scored automatically by using a custom tracking and control program (CRITTA)[92].

**Odor delivery and odor concentrations**. Odor delivery in the MOT was done as previously described[16], with some protocol modifications. The rack with stacked conditioning chambers was connected to an olfactometer that was used to deliver precisely timed odor stimuli (Supplementary Fig. 2b). The conditioning odors methylcyclohexanol (MCH) and 3-octanol (OCT) were carried by dry, compressed air and routed through mass flow controllers (MFC; Sensirion AG). Carrier air flow was controlled with two 2 L/min capacity MFCs and pushed through a humidifying gas washing bottle containing distilled water (Schott Duran) at 0.6 L/min. Odor streams were controlled with 500 mL/min MFCs and pushed through glass vials containing pure liquid odorants (either MCH or OCT, respectively). Prior to conditioning, the odor concentrations were adjusted to ensure that flies did not display a strong preference for one of the odors over the other prior to training. Odor administration was carried out with the following MFC settings: OCT left side 25–35 mL/min; OCT right side 30–40 mL/min; MCH left side 50–60 mL/min;

MCH right side 50–60 mL/min. Odor presentation at the behavioral chamber arms was switched with computer-controlled solenoid valves (The Lee Company, USA). The MFCs were regulated via CRITTA (LabView software). At ad hoc intervals between experiments, odor concentrations were measured with a photoionization detector (PID, RAE systems; PGM-7340). The experiments were performed with a relative concentration of 14–16 parts per million (ppm) for MCH and 6–8 ppm for OCT in the chambers. A relative humidity of 70–75% was maintained via regulation of the air flow; this was monitored (ad hoc, between experiments) by using a custom humidity sensor with a custom LabVIEW code (National Instruments, USA).

**Olfactory conditioning and data visualization.** Classical olfactory conditioning has been described previously[5,6,16]. Before each experiment, flies were briefly anesthetized on ice and six flies were loaded into each conditioning chamber (Supplementary Fig. 2A–B). Each conditioning experiment began with an acclimatization (baseline test) phase where *Drosophila* were exposed to both odors in the absence of a shock stimulus. Subsequently, in the first stage of training the chambers were flushed with carrier air and flies were exposed to either MCH or OCT in the presence of a shock stimulus (12 shocks at 60 V during a 60-s time interval). During the second stage of training the shock-paired odor was removed and the flies were exposed to the other odor in the absence of shock. After removal of the odor and air-puff agitation, flies were tested for shock-odor avoidance. The flies were given a choice between the two odors and average shock-odor avoidance was quantified for the last 30 s of the 2 min-long testing phase. The main stages of the conditioning protocol are summarized in Fig. 2b. The full conditioning protocol is presented in Supplementary Fig. 2d. The shock-odor avoidance of flies for each conditioning trial was expressed as a performance index (PI)[6]; however, instead of a single endpoint, counting was performed on individual video frames over the final 30 s of the testing period. Each trial produced a half PI against the respective conditioned odor (either MCH or OCT) and two half PI's from consecutive experiments (with different conditioning odors) were combined to a full PI (full PI = half PI OCT + half PI MCH). For data visualization, the distribution of full PI's was plotted with a 95% CI error presenting a ΔPI between control and test genotypes by using the DABEST estimation-statistics package[91].

**Behavioral data analysis.** Analyses of STM and survival experiments were performed with estimation statistics[93–95]. The rationale for the estimation framework was described previously[91,96]. For data analysis and visualization, the individual values of full PIs were plotted with standard-deviation lines; to describe the differences between control and test genotypes, the distributions of ΔPI with 95% confidence intervals were plotted with the DABEST estimation package (www.estimationstats.com). Using this approach, we avoid null-hypothesis significance testing[97,98] in favor of estimation plots, which show the relevant effect sizes and comprehensive distributional information of both observed and inferred values, focusing on the intervention effect size. For legacy purposes only, the two-group comparison permutation $P$ values are listed in the respective legends for each figure. The sample size of at least 72 flies per group for behavioral experiments was based on precision planning to accommodate an average target margin of error of 0.33 standardized effect-size units[94]. This is equivalent to 80% power to detect a 0.5 SD effect size (https://www.esci.thenewstatistics.com/esci-precision.html#tab-1).

***Drosophila* immunostaining.** We performed immunostaining as previously described[99], by fixing dissected tissues in PBS containing 4% formaldehyde and incubating with the following primary antibodies: mouse α-HA (1:1000, Santa Cruz), and guinea pig α-Mettl3 (1:2000, gift of Cintia Hongay, Clarkson University). Alexa Fluor-488, and -568 secondary antibodies were from Molecular Probes and used at 1:1000. Tissues were mounted in Vectashield mounting buffer with DAPI (Vector Laboratories). Images were captured with a Leica SP5 confocal microscope; endogenous GFP signals were monitored.

**m$^6$A individual-nucleotide-resolution cross-linking and immunoprecipitation (miCLIP).** miCLIP libraries were prepared by subjecting RNA samples to the established protocol[66] with the minor changes described below. Briefly, total RNA was collected from <1-week-old *w1118* (wild type) and *Mettl3[null]* (mutant) female heads using TRIzol RNA extraction. Poly(A) + RNA was enriched using two rounds of selection. RNAs were fragmented, incubated with α-m$^6$A (202 003 Synaptic Systems) and crosslinked twice in a Stratalinker 2400 (Stratagene) using 150 mJ/cm$^2$. Crosslinked RNAs were immunoprecipitated using Protein A/G magnetic beads (Thermo) and washed under high salt conditions to reduce non-specific binding. Samples were radiolabeled with T4 PNK (NEB), ligated to a 3′ adaptor using T4 RNA Ligase I (NEB), and purified using SDS–polyacrylamide gel electrophoresis (SDS–PAGE) and nitrocellulose membrane transfer. RNA fragments containing crosslinked antibody peptides were recovered from the membrane using proteinase K (Invitrogen) digestion.

Recovered fragments were subjected to library preparation. First-strand cDNA synthesis was performed using SuperScript III (Life Technologies) and iCLIP-barcoded primers, which contain complementary to the 3′ adaptor on the RNA. cDNAs were purified using denaturing PAGE purification, circularized using CircLigase II (EpiCentre), annealed to the iCLIP Cut Oligo, and digested using

BamHI (Thermo). To generate libraries for sequencing, the resulting linear cDNAs were amplified using Accuprime SuperMix I (Invitrogen) and P5 and P3 Solexa primers, and purified using Agencourt AMPure XP beads (Beckman Coulter).

For input libraries, poly(A) + RNAs were fragmented and directly subjected to radiolabelling and 3′ adaptor ligation. All subsequent steps are as listed above. Libraries were paired-end sequenced on an Illumina HiSeq2500 instrument at the New York Genome Center (NYGC).

**miCLIP bioinformatic analyses.** Read processing, mutation calling, and annotation of CIMs was performed as described[66]. Briefly, to prepare libraries for mapping, adapters and low-quality reads were trimmed using flexbar v2.5. Next, the FASTQ files were de-multiplexed using the pyBarcodeFilter.py script from the pyCRAC suite. Random barcodes were removed from sequencing reads and appended to sequence IDs using an awk script and PCR duplicates were removed using the pyCRAC pyDuplicateRemover.py script. Paired end reads were merged and mapped to the *Drosophila* reference genome sequence (BDGP Release 6/dm6) using Novoalign (Novocraft) with parameters –t 85 and −l 16.

Mutations were called using the CIMS software package[100]. To identify putative m$^6$A sites, C-to-T transitions with preceding A nucleotides were extracted and filtered such that the number of mutations that support the mismatch ($m$) > 1 and $0.01 < m/k < 0.5$, where $k$ is the number of unique tags that span the mismatch position.

Peaks were called by adapting the model-based analysis for ChIP-Seq (MACS) algorithm[101]. Mettl3-dependent peaks for head libraries were determined using miCLIP versus input, comparing wild type and *Mettl3* libraries and the MACS2 differential binding events program (bdgdiff) with parameters −g 20 and −l 120. Lastly, peaks were split using PeakSplitter (version 1.0, http://www.ebi.ac.uk/research/bertone/software).

To generate nucleotide content plots, filtered C-to-T transitions with preceding A nucleotide (as mentioned above) that mapped within the top 100 or 1000 Mettl3-dependent peaks were chosen to describe the nucleotide content surrounding CIMs. Sequences were obtained using the *Drosophila* reference genome sequence (dm6) and fed to WebLogo version 2.8.2 with the frequency setting[102].

Custom scripts were used to generate metagene plots. Briefly, to prepare mapped data, each miCLIP bam file was converted to bedGraph format with span of 1 nucleotide. To prepare features, for each gene, the longest transcript model was selected and divided into 5′ UTR, CDS and 3′ UTR segments according to Ensembl transcript models for BDGP6.94. Next, miCLIP read depth mapping to transcripts were selected and scaled such that each 5′ UTR, CDS and 3′ UTR were 200, 1000, and 300 nts. To normalize, the score at each scaled nucleotide was divided by the total score across all 1500 nucleotides. Finally, to yield metagene score across each feature (UTRs and CDS), genes of interest were selected and means were calculated for each nucleotide position. Smoothing functions from the ggplot2 package[103] were used to visualize metagene analysis.

Pie charts were obtained by mapping peaks to Ensembl transcript models for BDGP 6.94. Since transcript features occasionally overlap, the following order was used to bin peaks into different categories: other (not mapping transcript models), introns, start codons, 5′ UTRs, 3′ UTRs, and CDS. Finally, the ggplot2 function geom_bar was used to plot the accounted annotations into a pie chart.

Input normalized miCLIP tracks along with described peaks were used to generate heatmaps using deepTools2 functions computeMatrix and plotHeatmap[104].

**RNA-seq analysis.** Flies of the specified genotypes(*w;;Mettl3[null]/+*, *w;;Mettl3[null]/Mettl3[cat]*, *w;;Df(3 R)BSC461/ +* , *w;;Ythdf[NP3]/Df(3 R)BSC461*), were aged to 1 or 3 weeks at 25 ℃. Female heads were dissected and collected for total RNA extraction using TRIzol reagent. Sequencing libraries were prepared using the TruSeq Stranded Total RNA Sequencing Kit (Illumina) following the manufacturer's protocol. Sequencing was performed on a HiSeq 2500 System in paired end read mode, with 100 bases per read at the Integrated Genomics Operation (IGO) at Memorial Sloan Kettering Cancer Center.

RNA sequencing libraries were mapped to the *Drosophila* reference genome sequence (BDGP Release 6/dm6) using HISAT2[105] under the default settings. Gene counts were obtained by assigning and counting reads to the Ensembl transcript models for BDGP6.94 using Rsubread[106]. Differential gene expression analysis was performed with comparisons as listed in Supplementary Dataset 5 using the R package DESeq2[107] and applying a strict adjusted $p$-value cutoff of 0.05.

**CRISPR/Cas9 deletion of *Mettl3* in S2-S cells.** We used CRISPR/Cas9-mediated mutagenesis as described[67] to generate *Mettl3-KO* S2 cell lines. Guide RNA sequences are listed in Supplementary Dataset 6. We analyzed 11 candidate clonal lines obtained from subcloning of two initial low-complexity mixed cell populations, and kept the deletions #3-5 and #4-3 as described in Supplementary Fig. 11.

**m$^6$A-RIP-PCR and RIP-rtPCR.** We adapted a protocol from our recent study[55]. Plasmids of 5 μg were transfected into 6 × 10[6] S2 cells using Effectene (Qiagen) and incubated for 3 days. Cells were washed with PBS and lysed with IP lysis buffer (30 mM HEPES, pH 7.5, 150 mM KOAc, 2 mM Mg(OAc)$_2$, 5 mM DTT, 0.1% NP40) supplied with Complete, EDTA-free Protease Inhibitor and 40 U mL$^{-1}$ SUPERase•In RNase Inhibitor (Ambion) on ice for 30 min, followed by 2 × 10 min

centrifugation at 20,000×g at 4 °C. 10% of the cleared cell lysate were kept as input and the rest was incubated with 15 μL Dynabeads™ Protein G (Thermo Fisher,10004D) (with HA or GFP antibody) for 4 h at 4 °C. RNase I (Invitrogen, AM2294) was added to the sample for RNase treatment at 0.2 U final concentration. The beads were washed three times using IP lysis buffer and then resuspended in 100 mL lysis buffer. To elute RNA, the beads were mixed with 900 mL of Trizol, vortexed for 1 min and incubated at RT for 50 with rotation. RNA extracted and treated were Turbo DNase (Ambion) for 30 min before cDNA synthesis using SuperScript III (Life technology) with random hexamers. PCRs were done using fusion high-fidelity polymerase (ThermoFisher Scientific).

For m$^6$A-RIP-qPCR, the mRNAs were immunoprecipitated using α-m$^6$A according to the procedure shown above. The IP-mRNAs were then reverse transcribed and amplified following the same protocol. The enrichment of m$^6$A was quantified using qPCR as reported. The sequences of qPCR primers are listed in Supplementary Dataset 6.

**RNA degradation assay**. S2 cells were seeded as $3 \times 10^6$ cells per well. Actinomycin-D (Gibco, 11-805-017) was added to a final concentration of 5 μM, and cells were collected before or 5 h after adding actinomycin-D. Then the cells were processed as described in 'RT–qPCR', except that the data were normalized to the $t = 0$ time point.

**SUnSET assay and Western blotting**. For each assay, we incubated $3 \times 10[6]$ cells in 1 mL Schneider's medium including 10% FBS for 5 min at 25 °C with or without 5 μg/mL puromycin (Gibco™ Sterile Puromycin Dihydrochloride). Cells were then washed twice with cold PBS, and lysed with 100 μL lysis buffer (10 mM Tris–HCl, pH 7.5, 300 mM NaCl, 1 mM EDTA, 1% Triton X-100, Protease Inhibitor Cocktail Roche). The cell pellet was resuspended by pipetting and incubated on ice for 30 min, then centrifuged at 16,000×g for 10 min at 4 °C. Protein concentration was measured using Bio-Rad Protein Assay Dye (500–0006) and 2.5 μg proteins were separated on SDS–PAGE and transferred to Immobilon-P membranes. Membranes were blocked for 1 h in TBS containing 5% nonfat milk and 0.1% Tween-20, followed by incubation with mouse α-puromycin (1:1000) overnight at 4 °C. Appropriate secondary antibodies conjugated to HRP (Jackson) were used at 1:5000 for 1 h at room temperature, then visualized using chemiluminescence detection (Amersham ECL Prime Western Blotting Detection Reagent). Mouse α-puromycin (2A4, 1:1000) and mouse α-ß-tubulin (E7, 1:1000) were from DSHB; guinea pig α-Mettl3 (1:5000) was a gift from Cintia Hongay.

**Luciferase sensory assays**. To generate the pAc5.1-aqz-wt-luc reporter, we cloned the *aqz* 5′ UTR (chr3R:8,818,731–8,820,682) into the 5′ UTR pAc5.1-luc vector encoding Firefly Luciferase (KpnI/EcoRI). We then used site-directed mutagenesis (primers listed in Supplementary Dataset 6) to generate pAc5.1-aqz-M-luc (8820387A, 8820401A into T).

To assay luciferase reporter activity and mRNA levels, $10^6$ S2 cells were seeded per well of 12-well plate and transfected with 150 ng pAc5.1-Renilla, and 150 ng pAc5.1-aqz-wt-luc, or pAc5.1-aqz-M-luc constructs. Luciferase activities were measured 3 days after transfection using Dual Glo luciferase assay system (Promega) and Cytation5 (BioTek) from 100 μL cell from each well of the 12-well plate. We calculated the ratio between Firefly and Renilla luciferase activities. The remaining cells in the 12-well plate were processed to extract total RNA (DNase I digested) by TRIzol® reagent (Invitrogen), followed by RT-qPCR quantification. The level of *F-luc* mRNA was normalized by that of *R-luc* mRNA.

**Reporting summary**. Further information on research design is available in the Nature Research Reporting Summary linked to this article.

## Data availability

All transgenic strains and plasmids generated for this study are available upon request. All of the raw miCLIP and RNA-seq data generated in this study were deposited in NCBI-GEO under GSE147230. Source data are provided with this paper. All of the raw and summarized STM, locomotion, and survival data are available from the zenodo repository under the following DOI: 10.5281/zenodo.4446416. Source data are provided with this paper.

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

## Acknowledgements

We thank the Developmental Studies Hybridoma Bank (DSHB) and Cintia Hongay for antibodies, Jean-Yves Roignant, Matthias Soller and the Bloomington Drosophila Stock Center (BDSC) for fly stocks, Stefan Ameres for the S2-S line and advice on cell mutagenesis. Sara Zaccara and Samie Jaffrey provided helpful advice on generation of miCLIP libraries and discussion about m6A. A.C.-C. and S.O. were supported by grants from the Singapore Ministry of Education (MOE2013-T2-2-054 and MOE2017-T2-1-089) and by Duke-NUS Medical School. S.O. was supported by a Khoo Postdoctoral Fellowship Award (Duke-NUS-KPFA/2017/0015). Work in the REK lab was supported by R01-GM132189. W.D. was generously supported by the Edward C. Taylor 3rd Year Graduate Fellowship in Chemistry. Work in the ECL lab was supported by NIH grants R01-GM083300 and R01-NS083833, and by the MSK Core Grant P30-CA008748.

## Author contributions

L.K. generated all constructs and transgenic animals; constructed the miCLIP and RNA-seq libraries; performed SUnSET, RNA degradation, and reporter assays, and all disc and CNS immunostainings. S.O. generated methods and software, conducted all the behavioral assays, and helped draft the manuscript. B.J. performed all the bioinformatic analyses. E.S.P. generated *mettl3-KO* cell lines and conducted co-IP and RIP tests. W.D. analyzed m6A-YTH factor interactions, performed proteomic analyses, and quantified modified nucleosides. A.C.-C., R.E.K., and E.C.L. supervised experiments in their respective laboratories, obtained funding, helped interpret analyses, and wrote the manuscript with input from all other authors. E.C.L. and A.C.-C. conceived the project.

## Competing interests

The authors declare no competing interests.
