## [Peer Review File · Nature Communications]

Reviewers' Comments:

Reviewer #1:

Remarks to the Author:

In this manuscript, Kan et al investigate the role of epitranscriptomic modification on cellular function and behavior in the fruit fly. The manuscript leverages many different cutting-edge genetic tools to examine the effects of mutating M6A writers and readers. A significant findings is that M6A modification is required for olfactory learning, and there seems to be an age-dependent component to the induction of short term memories. In addition, the manuscript includes detailed analysis of mettl3-dependent m6A modifications that will provide a valuable resource to the community. The presented experiments are scientifically rigorous, and the paper well-written, though it could be improved by greater integration across the biological systems being studied, additional statistical analysis, and additional discussion of how m6a contributes to regulation of plasticity and memory. Overall, this manuscript represents a significant contribution to our understanding of the role of epitranscriptomics in memory formation. The specific concerns described below can likely be addressed without additional experiments (though they may be helpful, for example point 6), given that most of us are currently have limited access to our labs.

1. The novelty and impact of this manuscript is largely based on assigning a critical role for m6A modifications in neural plasticity and behavior. The introduction is long, with relatively little background on memory. Given the broad readership of the journal it would be helpful to streamline discussion of epitranscriptomics and introduce readers to memory in flies.
2. The conclusions relating to FMR1 are not clear from the text. While this is an attractive candidate given its role in memory (and other behaviors) and previous studies implicating FMRP/FMR1 as an M6A binding protein, this portion should be expanded on for clarity.
3. This is may be semantic, but referring to short-term memory as STM is appropriate, but generalizing it as 'memory' seems misleading. Given the nature of the assay, it may be better to refer to this as 'learning' in the discussion since the behavioral assay is likely reflective of the initial plasticity, rather than any form of memory consolidation. Was later memory assayed?
4. I do not understand why statistics are not applied to behavioral data. In this case a simple t-test would support differences that are clearly robust. As it is written, the authors use words like 'substantially' but it is not clear what this means without the statistical analysis.
5. More explanation is needed for the controls used to account for background strain. Trans-het or mutation over deficiency are not as powerful as outcrossing to an isogenic strain.
6. Figures 2 G and H show compelling evidence that mettl3 functions in neurons and then the mushroom bodies to regulate memory. One tests sufficiency, the other tests necessity. It would be helpful if to perform both experiments in each population e.g. rescue using mb247.
7. It would seem critical to show that disruption of Mettl3 does not impact olfaction or locomotion (i.e. they can perform the assay, and the defect is specific to memory).
8. Why are controls combined in figure 3E?
9. Figure 7 validates using the imaging disc, but it is unclear how this relates to experiments in S2 cells or the mushroom bodies. This can be addressed through discussion associating the systems examined within the manuscript.
10. In my opinion, the age-dependent effects on behavior are given insufficient discussion. Exactly how might m6A act within mushroom bodies to regulate learning in an age-dependent manner. One can infer M6A is not required for maintenance of neural function because longevity is

unaffected in a number of these situations, so is the idea it's required for plasticity? Or selectively within circuits encoding memory?

Reviewer #2:

Remarks to the Author:

In this manuscript Kan et al functionally and molecularly characterized the cytoplasmic YTH reader in *Drosophila melanogaster*. A few studies previously characterized the m6A pathway in this organism, in which an important function was uncovered in the sex determination pathway through splicing regulation of Sex lethal. This function is mediated through the nuclear YTHDC1. However no function was yet ascribed to YTHDF.

Here the authors demonstrated that the two YTH homologs are the main m6A readers in *Drosophila*, which parallels the findings in vertebrates. However they found distinct binding specificity for both readers. Since the loss of YTHDF2 is viable they went on to address the fly behavior. A specific defect in short-term memory in particular in older flies was identified. To determine potential targets m6A was mapped using fly head tissues and m6A enrichment was surprisingly found at 5' end of transcripts, which is in sharp contrast with the distribution in other species. Finally, they reported the absence of m6A impact on mRNA stability but rather found a potential function on translational activation.

At least two studies in mice already attributed roles for m6A in learning and memory, so in this sense this study is not conceptually novel. Nevertheless it is nice to see that this function is conserved in a different organism and I believe it would be a great addition to the previous reports. Evolutionary it is also an interesting model since as the authors pointed out the fly has only 2 YTH homologs, with only one expressed in the cytoplasm. In vertebrates it is still unclear if and why YTHDF proteins have distinct activities. Hence, the study in fly may help uncovering ancestral m6A cytoplasmic function.

This work is overall well done and the manuscript is clearly written. Nevertheless I feel that some (main) conclusions are not fully substantiated yet by the data and will certainly require additional work. Specifically, the effect of YTHDF on translation is not yet clear, and the molecular defect(s) that results in the behavior phenotype is currently a mystery. The latter point perhaps goes beyond the scope of this study but at least the effect on translation should be better supported.

Main comments:

1) One of the critical aspects of this work is to define the molecular role of YTHDF in *Drosophila*. As it stands this has not yet been properly addressed. Most experiments were performed in overexpression conditions, which might not recapitulate the real function of this protein. This is particularly important since the authors conclude that YTHDF binds poorly translated targets, although it is suggested that YTHDF enhances translation. This sounds rather counterintuitive (even though not impossible).

The effect of the loss of METTL3 on nascent translation (50% decrease) is surprising given that its absence is dispensable for fly viability. To validate this effect and rule out off target activity a rescue experiment should be carried out. Furthermore, does this decrease also occur *in vivo*? And does the YTHDF KO also reduce global nascent translation?

For the *in vivo* demonstration, the overall GFP intensity of panel 7B is increased compared to 7A, thus it is difficult to appreciate the specific increase in the *ptc* expression domain. It would be important to repeat this experiment, and also assess GFP expression under YTHDF loss of function. Furthermore since the exact m6A sites on *aqz* 5'UTR are known it would be a perfect control to mutate these sites and address the effect of YTHDF loss and gain of functions. Since generating transgenic lines is time consuming perhaps this experiment could be done in a cell culture system with luciferase reporters.

2) The part with Fmr1 is confusing. It is odd that the mutant 3A still interacts with Fmr1 while the treatment with RNase abolishes this interaction. Since this part doesn't add much to the characterization of YTHDF2 and is rather distracting I suggest to leave this out. If the authors want to keep it more work is required to understand how these two proteins interact and how they may influence each other function.

3) Based on previous work it is surprising that YTHDC binds so poorly m6A probes, in particular the GGACU (Sup Fig 1). Why were the binding experiments done in HEK cells and not in Drosophila cells? Drosophila YTH proteins can easily be competed with human factors with different affinity for m6A and therefore it's difficult to conclude about YTH differential binding. In fact this differential binding is not observed later in RIP experiment. YTHDF and YTHDC seem to bind indistinctly their targets in RIP assays. Therefore it's difficult to grasp the take home message from these experiments. Binding to GGACU should be repeated with Drosophila cell extracts using the full length YTHDC and YTHDF to get a clear answer about their differential binding behavior.

4) The authors used Mettl3 RNAi to test the cell autonomous behavior in the mushroom body (fig 2H). RNAi can have off target effect and without rescue experiments it's hard to conclude for specific effects. Expressing Mettl3 with the MB driver in Mettl3 null mutant (as they did for elav) would be a better experiment. The genotypes for combined controls are not indicated (also true for other figures)

5) Is the morphology of the mushroom body in METTL3 or YTH mutant proteins normal? In other words, is it a structural defect that yields the behavior phenotypes or is it a functional defect?

6) Figs 5B,C show distinct binding behavior for WT and mutant YTH protein. Does the association of YTH protein decrease upon Mettl3 KO? The authors mentioned that a binding pattern did not emerge possibly due to overexpression (which again highlights the difficulty of interpreting overexpression experiments). What are the binding sites for these few transcripts? This relates to point 3. Does it fit with the uncovered binding specificity in Fig 1?

Minor comments

L113. The phenotype of METTL3 KO in mouse is rather dramatic so the statement minimizing the impact of m6A should be tuned down.

Lanes 288-290. The two sentences are somewhat contradicting. The neural depletion of YTHDC on longevity by RNAi has only a mild effect so why the authors conclude that neural YTHDC has a major role in this context?

The color code for Sup Fig 6A-C is confusing. Please keep the same color for the same genotypes.

Lane 461: For the mRNA decay assay in fig 6f, were the targets validated for methylation?

Conclusion

This manuscript has many strengths, in particular the nice combination of phenotypic and molecular characterization. However, the study is somewhat weakened by the previous publications on the link between m6A and learning and memory functions. Nevertheless Drosophila offers a unique system to address YTHDF cytoplasmic function and in this sense I would have liked to see better characterization of this protein at the molecular level and its link to translation.

REVIEWER COMMENTS

Reviewer #1 (Remarks to the Author):

In this manuscript, Kan et al investigate the role of epitranscripomic modification on cellular function and behavior in the fruit fly. The manuscript leverages many different cutting-edge genetic tools to examine the effects of mutating M6A writers and readers. A significant findings is that M6A modification is required for olfactory learning, and there seems to be an age-dependent component to the induction of short term memories. In addition, the manuscript includes detailed analysis of mettl3-dependent m6A modifications that will provide a valuable resource to the community. The presented experiments are scientifically rigorous, and the paper well-written, though it could be improved by greater integration across the biological systems being studied, additional statistical analysis, and additional discussion of how m6a contributes to regulation of plasticity and memory. Overall, this manuscript represents a significant contribution to our understanding of the role of epitranscriptomics in memory formation. The specific concerns described below can likely be addressed without additional experiments (though they may be helpful, for example point 6), given that most of us are currently have limited access to our labs.

We thank the referee for their enthusiasm for our work.

1. The novelty and impact of this manuscript is largely based on assigning a critical role for m6A modifications in neural plasticity and behavior. The introduction is long, with relatively little background on memory. Given the broad readership of the journal it would be helpful to streamline discussion of epitranscriptomics and introduce readers to memory in flies.

We have completely rewritten the introduction with this in mind. We provide several new paragraphs to open the paper with the history of Drosophila learning and memory genetics and circuitry, to try to place our work into context.

2. The conclusions relating to FMR1 are not clear from the text. While this is an attractive candidate given its role in memory (and other behaviors) and previous studies implicating FMRP/FMR1 as an M6A binding protein, this portion should be expanded on for clarity.

We agree this is not very integrated. The referee will see that we have added extensive molecular tests, mechanistic tests, behavioral rescue assays and immunostaining assays of MB morphology that not only extend our story, but make the FMR1 tests even more peripheral. Given our study is now very long, we feel it is best to remove this part as it is supplemental only, since the other referee also suggested to cut these results.

3. This is may be semantic, but referring to short-term memory as STM is appropriate, but generalizing it as 'memory' seems misleading. Given the nature of the assay, it may be better to refer to this as 'learning' in the discussion since the behavioral assay is likely reflective of the initial plasticity, rather than any form of memory consolidation. Was later memory assayed?

We take the referee's point. We focused our behavioral assays on STM. Accordingly, we have clarified mention of 'memory' in the text to STM.

4. I do not understand why statistics are not applied to behavioral data. In this case a simple t-test would support differences that are clearly robust. As it is written, the authors use words like 'substantially' but it is not clear what this means without the statistical analysis.

Our lab (ACC) has published extensively on the topic of estimation statistics and the rationale for this analysis framework (Ho et al. 2019; Claridge-Chang and Assam 2016). To respond and clarify this rationale, we have added a paragraph to the methods section. For legacy purposes only, we also include the two-group comparison permutation P values are listed in the respective legends for each figure.

5. More explanation is needed for the controls used to account for background strain. Trans-het or mutation over deficiency are not as powerful as outcrossing to an isogenic strain.

We appreciate the referee's concern and we also seek to rule out background effects as much as possible, especially in behavioral assays that are prone to such confounds. While we also utilize outcrossing (e.g. Garaulet Dev Cell 2020), in our experience, analyzing trans-heterozygous and hemizygous allelic combinations are an appropriate means of ruling out most second-site mutations. The lack of memory defects in *ythdc1* mutants, generated in an identical genetic background by CRISPR, and which otherwise phenocopy *mettl3* in many overt respects, provides another argument against background strain effects.

However, the gold-standard is to provide a transgenic means of rescue. This is not always possible, especially when transgenes do not supply a precise level or localization of the gene product. Nevertheless, we also show both *mettl3* mutants and *ythdf* mutants were rescued by celltype-specific expression of the cognate gene products within whole-animal mutants. In fact, we extended these data to show that the requirement resides specifically within the mushroom body (see below). We believe this combination of genetic tests provides a strong foundation to conclude that m6A/YTHDF pathway has a neural-autonomous role in learning and memory in *Drosophila*.

6. Figures 2 G and H show compelling evidence that mettl3 functions in neurons and then the mushroom bodies to regulate memory. One tests sufficiency, the other tests necessity. It would be helpful if to perform both experiments in each population e.g. rescue using mb247.

Given the pandemic situation that we were locked out of the lab for 3 months, it has complicated efforts to construct the requisite recombinant stocks, verify and amplify them, and then to conduct the learning and memory tests, which have to be done in aged flies. We note that each step had to be done perfectly and dovetail with no delays.

We are pleased that we got some very interesting results. In fact, while it is undoubtedly the case that m6A plays regulatory roles in non-neuronal cells, and even in neurons outside of the mushroom body (MB), we were able to rescue the memory defects of *mettl3* knockout animals by re-expressing *Mettl3* in the MB. We also obtained substantial rescue of *ythdf* knockout animals by re-expressing *YTHDF* only in the MB. These sufficiency experiments are striking and the data were added to Figure 7.

In response to Referee 2, we also performed a set of immunostaining studies that reveal a striking structural defect in MB organization in both *mettl3* and *ythdf* mutants. These data were also added to Figure 7.

Given these revisions, we re-organized the paper and moved all the MB data into Figure 7.

7. It would seem critical to show that disruption of Mettl3 does not impact olfaction or locomotion (i.e. they can perform the assay, and the defect is specific to memory).

To address the referee's concern, we examined the olfactory acuity and shock sensitivity of m6A mutants, and found that they were comparable to the w1118 control. We also looked at the locomotor response of w1118 and m6A mutants before, during and after a shock stimulus. While the speed of the flies does vary slightly before the shock onset, the locomotor response of the flies across the different genotypes looks very similar. This suggests that the sensitivity to shock in m6A mutants is not impaired (the spikes in speed correspond to each respective shock stimulus i.e. 12 shocks in 60sec).

To test whether olfactory acuity is comparable between m6A mutants and the w1118 control stock we compared MCH avoidance scores when flies were given a choice between the MCH odor and clean air in the absence of conditioning, testing different MCH concentrations. The avoidance score between the w1118 control and the m6A mutants remains comparable- suggesting that (at least for MCH) the olfactory acuity of the flies is similar.

We now include these panels into a new supplementary figure (S6).

8. Why are controls combined in figure 3E?

We have now separated the driver and responder controls as requested.

9. Figure 7 validates using the imaging disc, but it is unclear how this relates to experiments in S2 cells or the mushroom bodies. This can be addressed through discussion associating the systems examined within the manuscript.

We mention in the text that the flat planar preparation of the wing imaginal disc epithelium makes it an ideal *in vivo* setting to test the regulatory impact of localized expression of YTHDF, with non-Gal4-expressing territories side-by-side as negative controls. We have used this assay in many other studies of post-transcriptional regulatory biology (e.g. miRNA studies) and found it to be sensitive. The 3-D structure of the CNS makes it difficult to make the same conclusions.

10. In my opinion, the age-dependent effects on behavior are given insufficient discussion. Exactly how might m6A act within mushroom bodies to regulate learning in an age-dependent manner. One can infer M6A is not required for maintenance of neural function because longevity is unaffected in a number of these situations, so is the idea it's required for plasticity? Or selectively within circuits encoding memory?

The aging aspect is interesting and deserves future attention. In our revision, we have conducted a definitive rescue experiment and shown that the short term memory defects in *mettl3* whole animal knockouts are fully rescued by re-expression of *Mettl3* only in mushroom body neurons, and that the same intervention with *YTHDF* can substantially rescue *ythdf* whole animal knockouts. This shows that even if there are contributions of m6A/*YTHDF* to other neurons, which seems likely, the MB is the most important for mediating the role of this pathway in this memory paradigm. We also add data that there are specific structural defects in MB in *mettl3/ythdf*, which is consistent with this idea.

However, a curious finding is that the MB β -lobe fusions manifest even earlier (1 week) than the behavioral defect in memory. We observe this in both *mettl3* and *ythdf* mutants. We don't know the connection, whether these are parallel effects, or the structural defect is a more sensitive phenotype than the behavioral defect. This requires further study, but our current study opens up many new pursuits.

Reviewer #2 (Remarks to the Author):

In this manuscript Kan et al functionally and molecularly characterized the cytoplasmic YTH reader in Drosophila melanogaster. A few studies previously characterized the m6A pathway in this organism, in which an important function was uncovered in the sex determination pathway through splicing regulation of Sex lethal. This function is mediated through the nuclear YTHDC1. However no function was yet ascribed to YTHDF.

Here the authors demonstrated that the two YTH homologs are the main m6A readers in Drosophila, which parallels the findings in vertebrates. However they found distinct binding specificity for both readers. Since the loss of YTHDF2 is viable they went on to address the fly behavior. A specific defect in short-term memory in particular in older flies was identified. To determine potential targets m6A was mapped using fly head tissues and m6A enrichment was surprisingly found at 5' end of transcripts, which is in sharp contrast with the distribution in other species. Finally, they reported the absence of m6A impact on mRNA stability but rather found a potential function on translational activation.

At least two studies in mice already attributed roles for m6A in learning and memory, so in this sense this study is not conceptually novel. Nevertheless it is nice to see that this function is conserved in a different organism and I believe it would be a great addition to the previous reports. Evolutionary it is also an interesting model since as the authors pointed out the fly has only 2 YTH homologs, with only one expressed in the cytoplasm. In vertebrates it is still unclear if and why YTHDF proteins have distinct activities. Hence, the study in fly may help uncovering ancestral m6A cytoplasmic function.

This work is overall well done and the manuscript is clearly written. Nevertheless I feel that some (main) conclusions are not fully substantiated yet by the data and will certainly require additional work. Specifically, the effect of YTHDF on translation is not yet clear, and the molecular defect(s) that results in the behavior phenotype is currently a mystery. The latter point perhaps goes beyond the scope of this study but at least the effect on translation should be better supported.

We thank the referee for their overall interest in our study, and have undertaken revisions to address their main concerns.

Main comments:

1) One of the critical aspects of this work is to define the molecular role of YTHDF in drosophila. As it stands this has not yet been properly addressed. Most experiments were performed in overexpression conditions, which might not recapitulate the real function of this protein. This is particularly important since the authors conclude that YTHDF binds poorly translated targets, although it is suggested that YTHDF enhances translation. This sounds rather counterintuitive (even though not impossible).

We appreciate the referee's concern for further mechanistic understanding. We have conducted many additional experiments, as detailed below. At the same time, we hope that it is placed into context that instead of being the 50th or more paper on mammalian YTHDF proteins, this is the very first study on the fly ortholog, and we have elucidated not only its genetic requirement in behavior, but also put it into epistatic context with the m6A pathway and spatial context within the nervous system (ie, the mushroom body). We complement this with the first tissue-specific m6A map in this organism as well as data on the regulatory impacts of m6A/YTHDF. We acknowledge that more can always be done, but we hope it is also acknowledged that this study is a very large advance on the Drosophila m6A field.

The effect of the loss of METTL3 on nascent translation (50% decrease) is surprising given that its absence is dispensable for fly viability. To validate this effect and rule out off target activity a rescue experiment should be carried out. Furthermore, does this decrease also occur in vivo? And does the YTHDF KO also reduce global nascent translation?

It appears that in vivo, flies are able to compensate for the loss of the m6A pathway, since null mutants of the pathway are viable. To us, this is one of the more surprising findings from model organism studies (e.g. Drosophila), considering the extremely broad and wide-ranging gene regulatory programs that have been associated with the mammalian m6A pathway, mostly from cell culture studies.

We attempted to conduct the puromycin incorporation assay in imaginal discs bearing clonal knockdown of mettl3. However, we did not observe overt changes in puromycin incorporation. At this point, we are not certain how to interpret this, whether it is truly a different result from S2 cells or whether we are not sufficiently expert in this assay using in vivo materials. We tried it a few times using different timecourses of puromycin labeling, but did not see obvious changes. For this reason, we cannot interpret YTHDF-KO by the same assay in vivo.

Our data in the mettl3-KO cell model indicate that the protein loss is most visible in the newly-synthesized pool, and/or that cultured cells are especially sensitive to observe this effect. To gain supporting evidence that this effect is genuine, we examined an independent mettl3-KO cell that we obtained through a separate population of knockout progenitors. This cell line has a larger deletion than the one we carried through with biochemical characterization, and also fails to accumulate Mettl3 protein by Western blotting. We did not initially characterize it because we later realized that it retains an

internal amplicon that potentially reflects a complex rearrangement within the *mettl3* locus. We conducted puromycin incorporation assays with this line, and observed that it also exhibits decreased nascent translation. We conducted these assays several times and provide quantification.

The data and characterization of this line are provided in main Figure 6 and Supplementary Figure 11. These findings corroborate our initial submission and we feel will be important for the field to know.

*For the in vivo demonstration, the overall GFP intensity of panel 7B is increased compared to 7A, thus it is difficult to appreciate the specific increase in the *ptc* expression domain. It would be important to repeat this experiment, and also assess GFP expression under YTHDF loss of function. Furthermore since the exact m6A sites on *aqz* 5'UTR are known it would be a perfect control to mutate these sites and address the effect of YTHDF loss and gain of functions. Since generating transgenic lines is time consuming perhaps this experiment could be done in a cell culture system with luciferase reporters.*

We conducted additional immunostaining experiments on discs and provide quantification of the numbers of discs analyzed. We are quite certain of the results, as even though the increase in GFP is modest, it is very specific to the cognate combination of wt-YTHDF on the m6A-*aqz* 5'UTR-tub-GFP reporter. In addition, there is an ideal negative control tissue built into the experiment, in addition to the non-cognate genetic combinations and the m6A-binding mutant YTHDF-3A transgene we tested.

We have also conducted an additional set of reporter tests using the *aqz* target as suggested. *aqz* 5' UTR is one of the most strongly modified targets in the genome, and we took the opportunity to compare a wt *aqz* 5'UTR-luciferase reporter with one in which we mutated two m6A binding sites present in the strongest Mettl3-dependent miCLIP peak. When tested in S2-S cells and normalized with a co-transfected renilla control, the *aqz*-wt reporter had about twice as much activity as the *aqz*-mutant, consistent with the notion that the m6A 5' UTR sites provide a stimulatory role. When we tested these in *mettl3*-KO cells, the wt and mutant *aqz* reporters had the same activities. These data support our original data and have been added to Figure 6.

2) The part with Fmr1 is confusing. It is odd that the mutant 3A still interacts with Fmr1 while the treatment with RNase abolishes this interaction. Since this part doesn't add much to the characterization of YTHDF2 and is rather distracting I suggest to leave this out. If the authors want to keep it more work is required to understand how these two proteins interact and how they may influence each other function.

Based on feedback of referee 1 and referee 2 that the Supplementary Fmr1 data are not well-integrated, along with the fact that we have added many new main Figure data panels, we have decided to leave the Fmr1 data out in revision, as suggested.

3) Based on previous work it is surprising that YTHDC binds so poorly m6A probes, in particular the GGACU (Sup Fig 1). Why were the binding experiments done in HEK cells

and not in Drosophila cells? Drosophila YTH proteins can easily be competed with human factors with different affinity for m6A and therefore it's difficult to conclude about YTH differential binding. In fact this differential binding is not observed later in RIP experiment. YTHDF and YTHDC seem to bind indistinctly their targets in RIP assays. Therefore it's difficult to grasp the take home message from these experiments. Binding to GGACU should be repeated with Drosophila cell extracts using the full length YTHDC and YTHDF to get a clear answer about their differential binding behavior.

We apologize that we might not have described the methods with sufficient clarity. The binding assays are all done in vitro using diazirine photoactivable RNA probes and immunoprecipitated proteins. In the tests shown, the proteins themselves were immunoprecipitated from HEK293 cells. The reason for this was that we initially found that we could not obtain sufficient amounts of the full-length YTH proteins when expressing them in S2 cells. We therefore cloned them into mammalian expression vectors and found that we could obtain the requisite proteins needed from HEK293 cells. We do not think it is likely that the cell source plays a major role in the RNA affinity of the proteins, since we are using metazoan cells. For comparison, many binding studies and crystallographic studies of YTH domain proteins utilize bacterially-expressed proteins). We have clarified the technical reasons about switching from S2-expressed proteins to HEK293-expressed proteins in the methods.

4) The authors used Mettl3 RNAi to test the cell autonomous behavior in the mushroom body (fig 2H). RNAi can have off target effect and without rescue experiments it's hard to conclude for specific effects. Expressing Mettl3 with the MB driver in Mettl3 null mutant (as they did for elav) would be a better experiment. The genotypes for combined controls are not indicated (also true for other figures)

We have separated out the control genotypes and plotted them separately.

We appreciate that RNAi can have off-target effects. That is why we also conducted rescue experiments, which we showed for neuronal (elav-Gal4) rescue of the cognate whole animal knockouts.

Given the pandemic situation that we were locked out of the lab for 3 months, it has complicated efforts to construct the requisite recombinant stocks, verify and amplify them, and then to conduct the learning and memory tests, which have to be done in aged flies. We note that each step had to be done perfectly and dovetail with no delays.

We are pleased that we got some very interesting results. In fact, while it is undoubtedly the case that m6A plays regulatory roles in non-neuronal cells, and even in neurons outside of the mushroom body (MB), we were able to rescue the memory defects of mettl3 knockout animals by re-expressing Mettl3 in the MB. We also obtained substantial rescue of ythdf knockout animals by re-expressing YTHDF only in the MB. These sufficiency experiments are striking and the data were added to Figure 7.

5) Is the morphology of the mushroom body in METTL3 or YTH mutant proteins normal? In other words, is it a structural defect that yields the behavior phenotypes or is it a functional defect?

We have conducted a new set of immunostaining experiments on *mettl3* and *ythdf* mutants. Indeed, we observed a high-frequency fusion of MB β -lobes in *mettl3* mutants and in *ythdf* mutants. Thus, there seems to be a good correlation of structural and functional defects.

Curiously, we also observed that these specific structural defects are manifest earlier (ie at both 1 week and 3 weeks) than the behavioral defects (no or only weak phenotype at 1 week, stronger defect at 3 weeks). Thus, we do not know whether these are parallel effects, or if the MB structure precedes or is more sensitive than cognitive defects. These experiments also do not rule out that there might not be additional functional defects, ie in neural physiology, but we find the data quite interesting and have been added to Figure 7.

6) Figs 5B,C show distinct binding behavior for WT and mutant YTH protein. Does the association of YTH protein decrease upon Mettl3 KO? The authors mentioned that a binding pattern did not emerge possibly due to overexpression (which again highlights the difficulty of interpreting overexpression experiments). What are the binding sites for these few transcripts? This relates to point 3. Does it fit with the uncovered binding specificity in Fig 1?

The analysis of YTH binding patterns in *mettl3*-KO is an interesting point, but we believe beyond the scope of this study. What we have shown is that *Drosophila* YTH domain proteins (YTHDC and YTHDF) both use their YTH domain to recognize m6A specifically, and that point mutants of this domain (W3A) reduce their association to m6A bearing targets. However, since YTH proteins might still interact with other RNA binding proteins (e.g. FMR1), they might still conceivably associate with other transcripts in *mettl3*-KO. This would need to be addressed in other studies.

Minor comments

L113. The phenotype of METTL3 KO in mouse is rather dramatic so the statement minimizing the impact of m6A should be tuned down.

We have rewritten the introduction, as suggested by Referee 1 to include more extensive consideration of the genetics and anatomy of *Drosophila* learning and memory. Therefore, we have largely removed the paragraph that included this sentence.

Lanes 288-290. The two sentences are somewhat contradicting. The neural depletion of YTHDC on longevity by RNAi has only a mild effect so why the authors conclude that neural YTHDC has a major role in this context?

We edited this to say "overt" instead of "major", nervous-system effects on longevity.

The color code for Sup Fig 6A-C is confusing. Please keep the same color for the same genotypes.

The color codes were unified for the same genotypes.

Lane 461: For the mRNA decay assay in fig 6f, were the targets validated for methylation?

We updated the figure to include genes validated for methylation and to use the same negative control genes used elsewhere.

Conclusion

This manuscript has many strengths, in particular the nice combination of phenotypic and molecular characterization. However, the study is somewhat weakened by the previous publications on the link between m6A and learning and memory functions. Nevertheless Drosophila offers a unique system to address YTHDF cytoplasmic function and in this sense I would have liked to see better characterization of this protein at the molecular level and its link to translation.

We hope the referee will view our extensive efforts to revise the manuscript during these challenging times, and recognize the importance of raising a new model system to integrate behavioral, genomic, and mechanistic analyses of the m6A pathway.

*** See Nature Research's author and referees' website at www.nature.com/authors for information about policies, services and author benefits.*

COVID 19 and impact on peer review

As a result of the significant disruption that is being caused by the COVID-19 pandemic we are very aware that many researchers will have difficulty in meeting the timelines associated with our peer review process during normal times. Please do let us know if you need additional time. Our systems will continue to remind you of the original timelines but we intend to be highly flexible at this time.

Reviewers' Comments:

Reviewer #1:

Remarks to the Author:

In the revised manuscript, the authors have addressed all my concerns from the initial round of review. While my review of the first submission was positive, the revised version is significantly improved through inclusion of additional data and text changes. These include a revised introduction that places the work in the in the context of memory. The revised version includes clarification for statistical analyses and critical data showing that *mettle3* and *ythdf* mutants have normal olfactory acuity. Overall this manuscript will establish flies as a model for investigating the effects of m6A modification on plasticity, and therefore represents a significant contribution to the field.

Reviewer #2:

Remarks to the Author:

The authors substantially revised their manuscript, in particular with respect to the contribution of m6A and Ythdf in translation. I am in favour of publication but I will appreciate if the authors can answer/clarify one last concern regarding the new Fig 6K. Given the previously reported effect on m6A on mRNA level it is critical to show that the direct effect of m6A occurs at the translational effect and not at the mRNA level. Fig 6 seems to indicate that this is the case but these are only correlations. What is the RNA level of the luciferase transcripts? Does the RNA level of *aqz-M-Luc* is similar to *aqz-wt-Luc* and thus doesn't account for the reduced luciferase activity?

Minor points:

There are some issues with the nomenclature. *Drosophila* proteins should be written *Mettl3*, *Ythdf* and *Ythdc1*. Genes names as well with first capital letter but in italics

Supplementary Table 4 is not cited in the text

The lack of effect on speed is surprising given the previous reports by Soller/Roignant's groups. Perhaps the authors should add a sentence to comment on this.

S2-S cells are introduced in L340 but are described only in L404.

L389. Previous work didn't show a stabilizing effect of m6A as only steady state level was examined. The effect on mRNA level could also be transcriptional. Please revise accordingly.

REVIEWER COMMENTS

Reviewer #1 (Remarks to the Author):

*In the revised manuscript, the authors have addressed all my concerns from the initial round of review. While my review of the first submission was positive, the revised version is significantly improved through inclusion of additional data and text changes. These include a revised introduction that places the work in the in the context of memory. The revised version includes clarification for statistical analyses and critical data showing that *mettl3* and *ythdf* mutants have normal olfactory acuity. Overall this manuscript will establish flies as a model for investigating the effects of m6A modification on plasticity, and therefore represents a significant contribution to the field.*

We are very glad that the referee is now fully in support of our study.

Reviewer #2 (Remarks to the Author):

The authors substantially revised their manuscript, in particular with respect to the contribution of m6A and Ythdf in translation. I am in favour of publication but I will appreciate if the authors can answer/clarify one last concern regarding the new Fig 6K. Given the previously reported effect on m6A on mRNA level it is critical to show that the direct effect of m6A occurs at the translational effect and not at the mRNA level. Fig 6 seems to indicate that this is the case but these are only correlations. What is the RNA level of the luciferase transcripts? Does the RNA level of aqz-M-Luc is similar to aqz-wt-Luc and thus doesn't account for the reduced luciferase activity?

We have conducted several additional tests of the luciferase sensors, comparing both luciferase activity and luciferase transcripts. With new tests, it remains clear that the luciferase activity of aqz-wt-Luc is higher than the aqz-M-Luc bearing the mutations of m6A sites, although slightly less of difference than previously (we now report 30% difference). Importantly, under improved conditions we now find that the level of both reporters is similar in *mettl3*-KO cells, and comparable to aqz-M-luc in wt S2-S cells. We now find that that the levels of aqz-wt-Luc and aqz-M-luc transcripts are similar, although the variation in qPCR is greater than what we observe with luciferase activity. Since only the luciferase activities are significantly different in wt cells, and we did not observe significance for mRNA levels, we believe this is consistent with a translational effect. We added this data to Figure 6, but also are careful not to overinterpret these data from one model reporter in transfection assays.

Minor points:

*There are some issues with the nomenclature. Drosophila proteins should be written *Mettl3*, *Ythdf* and *Ythdc1*. Genes names as well with first capital letter but in italics*

We have updated the nomenclature as suggested.

Supplementary Table 4 is not cited in the text

We included a citation for Table S4.

The lack of effect on speed is surprising given the previous reports by Soller/Roignant's groups. Perhaps the authors should add a sentence to comment on this.

The assays used are fundamentally different. The previous reports mentioned used climbing assays and a locomotion assay known as Buridan's paradigm; we also documented a climbing defect in our own study (Kan Nat Comm 2017). In our MOT assay, the walking speed of flies is very different from activity measurements in either of these other assays. E.g. in a climbing assay flies experience, arousal, gravitation and other stimuli that are not as such present in the MOT setup. Furthermore, in a Buridan's paradigm flies are walking freely, usually between two black stripes. To prevent them from escaping their wings are cut which might also increase their locomotion because the flies may be stressed.

When we look further into the locomotion values, in the MOT we routinely observe an average locomotor activity of about 2mm/sec in wild type flies in the absence of agitation. In Lence et al the authors report much higher locomotion speeds (about 15mm/sec, fig 3C in the Nature paper). We don't think that fly speed is abnormally low in the MOT, its more likely that fly speed calculated in the Buridan is abnormally high. From our experience and literature findings, flies appear to actually move more when spatially restricted. When we consulted further with experts in Buridan's paradigm, we learned that it is typical to eliminate all resting periods in calculations of speed. This probably accounts for the 8x higher locomotion speed measured in the Nature paper, but its unclear if they are losing relevant parameters because different flies could be experiencing different levels of stationary phase.

In any case, we can be confident that our measurements detect relevant locomotion changes in the MOT, since flies undergo a reproducible increase in movement when agitated; however, this is similar between heterozygote and mutant *Mettl3* animals. Overall, we don't think its very important to dwell on these differences in what is a very lengthy paper. We will take the referee's suggestion as a brief edit, to cite that other literature noted a locomotion defect using a very different behavioral assay, which might be a potential confound for the memory performance. However, in the parameters of the MOT assay, there is no substantial difference recorded, so it will not affect our conclusions regarding memory.

S2-S cells are introduced in L340 but are described only in L404.

We moved the description of S2-S cell earlier.

L389. Previous work didn't show a stabilizing effect of m6A as only steady state level was examined. The effect on mRNA level could also be transcriptional. Please revise accordingly.

We state that the previous study suggested that m6A was correlated with slightly increased steady state levels of target mRNAs.

Reviewers' Comments:

Reviewer #2:

Remarks to the Author:

The authors have addressed all my last concerns and suggestions. The findings of this study are important for the m6A field and should provide new avenues for future research directions.